# Ancestral reconstruction reveals mechanisms of ERK regulatory evolution

Dajun Sang[1], Sudarshan Pinglay[1], Rafal P Wiewiora[2,3], Myvizhi E Selvan[4,5], Hua Jane Lou[6], John D Chodera[2], Benjamin E Turk[6], Zeynep H Gümüş[4,5], Liam J Holt[1]*

[1]Institute for Systems Genetics, New York University Langone Medical Center, New York, United States; [2]Sloan Kettering Institute, Memorial Sloan Kettering Cancer Center, New York, United States; [3]Memorial Sloan Kettering Cancer Center, New York, United States; [4]Department of Genetics and Genomic Sciences, Icahn School of Medicine at Mount Sinai, New York, United States; [5]Icahn Institute for Data Science and Genomic Technology, Icahn School of Medicine at Mount Sinai, New York, United States; [6]Department of Pharmacology, Yale University School of Medicine, New Haven, United States

**Abstract** Protein kinases are crucial to coordinate cellular decisions and therefore their activities are strictly regulated. Previously we used ancestral reconstruction to determine how CMGC group kinase specificity evolved (Howard et al., 2014). In the present study, we reconstructed ancestral kinases to study the evolution of regulation, from the inferred ancestor of CDKs and MAPKs, to modern ERKs. Kinases switched from high to low autophosphorylation activity at the transition to the inferred ancestor of ERKs 1 and 2. Two synergistic amino acid changes were sufficient to induce this change: shortening of the β3-αC loop and mutation of the gatekeeper residue. Restoring these two mutations to their inferred ancestral state led to a loss of dependence of modern ERKs 1 and 2 on the upstream activating kinase MEK in human cells. Our results shed light on the evolutionary mechanisms that led to the tight regulation of a kinase that is central in development and disease.

*For correspondence:
liam.holt@nyumc.org

Competing interests: The authors declare that no competing interests exist.

## Introduction

Protein kinases regulate most aspects of *Eukaryotic* biology, including cell growth, differentiation, proliferation and apoptosis. They are one of the largest protein families in eukaryotes (*Lahiry et al., 2010*); more than 500 kinases in the human genome comprise ~2% of all protein coding genes. Due to their central importance in signaling networks, kinase activity has evolved to be precisely regulated by a myriad of distinct mechanisms. A breakdown in kinase regulation is the underlying cause of many human diseases; in particular, inappropriate kinase activation drives most cancers.

Kinases have evolved a multitude of mechanisms to control their activation. Some examples such as the requirement for allosteric modulators, regulation by inhibitory interactions, and localization control are highly variable across the kinome. However, the activation of most kinases requires phosphorylation of the activation loop, a conserved structural feature that connects the N and C-terminal lobes of the kinase domain. Most serine/threonine kinases and a subset of tyrosine kinases are regulated by phosphorylation of a conserved TxY motif within this loop. This phosphorylation affects both substrate binding and the arrangement of catalytic residues (*Johnson et al., 1996*; *Nolen et al., 2004*), leading to robust regulation of enzyme activity.

Extensive structural and biochemical studies have afforded insight into the mechanisms of kinase regulation by activation loop phosphorylation. In the mitogen activated protein kinases (MAPKs) extracellular signal-regulated kinase 1 (ERK1) and ERK2, several structural elements are important for activity. These include the p+1 binding pocket that binds the substrate peptide proline, the

regulatory αC helix, and the L16 loop. Upon phosphorylation of the TxY threonine and tyrosine residues, a number of changes occur. A network of electrostatic contacts with the phospho-threonine and phospho-tyrosine drives a conformation change around the p+1 substrate binding site, enabling substrate binding. The regulatory αC helix rotates inwards, allowing a glutamate on the αC helix (Glu-69) to form a salt bridge with Lysine (Lys-52) on the β3 strand that is crucial for catalytic activity. Extension of the C terminal L16 loop enables rotation of the N- and C-terminal domains, leading to 'domain closure', bringing Lys-52 closer to other catalytic residues, such as aspartates in the DFG and HRD motifs. Together, these conformational changes can lead to a 1000-fold increase in activity (*Zhang et al., 1994*; *Canagarajah et al., 1997*; *Roskoski, 2012*). Similar conformational changes contribute to the regulation of many kinases beyond the MAPK family (*Huse and Kuriyan, 2002*; *Taylor and Kornev, 2011*).

Some kinases are able to autophosphorylate and thus autoactivate. Multiple mechanisms for autophosphorylation have been proposed (*Beenstock et al., 2016*). Some kinases, for example most receptor tyrosine kinases (*Beenstock et al., 2016*) are autophosphorylated in *trans*, where distinct molecules transfer phosphates to one another. Other kinases autophosphorylate in *cis;* that is the kinase catalyzes an intramolecular transfer of phosphates onto its own TxY. In the latter case, autophosphorylation of dual-specificity tyrosine regulated kinases (DYRK) and glycogen synthase kinase-3β(GSK3β) has been proposed to occur during intermediate stages of protein folding (*Lochhead et al., 2005*). Other mechanisms include binding to allosteric modulators that reorient the activation loop towards catalytic residues (e.g. p38α, *DeNicola et al., 2013*) or conformational changes driven by dimerization (e.g. RAF; *Rajakulendran et al., 2009*). In contrast, many other kinases require phosphorylation of the activation loop in *trans* by a distinct upstream kinase. For example, many MAPKs are dependent on phosphorylation by upstream MAPK kinases (MAPKKs) for their activation (*Chang and Karin, 2001*).

The MAP kinases ERK1 and ERK2 exhibit very low autophosphorylation activity as purified proteins in vitro (*Levin-Salomon et al., 2008*; *Beenstock et al., 2014*), and they manifest high activity in vivo only when they are phosphorylated by the upstream MAPKK, MAP/ERK kinase (MEK). ERK1 and ERK2 mutants have been described that exhibit much higher autophosphorylation activity (*Emrick et al., 2006*), and one such mutant can drive oncogenic transformation of cells (*Smorodinsky-Atias et al., 2016*) highlighting the importance of stringent control of ERK phosphorylation.

In earlier work (*Howard et al., 2014*), we used ancestral reconstruction methods to study the evolution of the specificity of the CMGC group of kinases, which include cyclin dependent kinase (CDK), MAPK, GSK, and casein kinase (CK). CDKs are major regulators of cell division (*Morgan, 2007*) and transcription (*Fisher, 2005*). MAPKs are crucial for cell proliferation, differentiation, and stress responses. As part of this study, we observed that an inferred deep ancestor of this group had high basal activity even in the absence of any activating kinase or allosteric activators. This finding was surprising given the tight control of many MAP kinases by upstream activators, and the dependence of CDKs on both cyclin binding and upstream activating kinases. Therefore, to study how suppression of autophosphorylation and thus strong dependence on upstream activating kinases evolved, we decided to investigate the regulatory evolution of the CMGC group, ultimately focusing on the MAP kinases ERK1 and ERK2.

Throughout the evolutionary history of MAP kinases, gene duplications and diversifications resulted in multiple paralogs with diverse regulation (*Howard et al., 2014*). For example, within the p38 subfamilies, p38β has a much higher basal catalytic activity than p38α (*Beenstock et al., 2014*) and within the ERK family, ERK7 is constitutively active (*Abe et al., 1999*; *Abe et al., 2001*), and the core kinase domain of ERK5 has a much higher autophosphorylation rate, and thus higher basal activity, than ERK1 and ERK2, which are almost completely inactive in the absence of the activating kinase MEK (*Buschbeck and Ullrich, 2005*).

Through biochemical analysis of reconstructed kinases, we found that the transition to dependence on MEK occurred between the inferred common ancestor of ERK1, 2 and 5, and that of ERK1 and 2. Through a targeted mutagenesis approach we identified the key evolutionary mutations that mediated this transition. Two changes— mutation of the gatekeeper residue and a single amino acid deletion in the linker loop between the β3 strand and αC helix accounted for this regulatory

evolution. Reversal of these inferred evolutionary mutations in modern ERK1 and ERK2 led to full autoactivation in the absence of an upstream activating kinase. Thus, we believe we have identified key steps in the regulatory evolution of the ERK MAPK family.

## Results

### Reconstruction of the lineage leading to ERK kinases

To understand how kinase activity evolved, we reconstructed ancestors of the kinases leading to ERK1 and ERK2. Briefly, we collected a set of sequences for each of the major kinase families of the CMGC group. The sequences sampled for each kinase family were selected to be evenly distributed across the eukaryotic phylogeny. We then used a maximum likelihood phylogenetic method (*Hanson-Smith and Johnson, 2016*) to infer the sequences of the ancestral kinases from AncCDK.MAPK from which all CDK and MAPK descend, to the inferred common ancestors of the MAPK (AncMAPK), Jnk/p38/ERK (AncJPE) and ERK families (AncERK, (*Howard et al., 2014*)). Posterior probabilities of the reconstructions of these four ancestral kinases are shown in *Figure 1—figure supplement 1*. In this study, we additionally generated a more focused reconstruction of ERK family kinases that reconstruct ancestors of ERKs including the inferred ancestor of ERK 1,2 and 5 (AncERK1-5), and that of ERK 1 and 2 (AncERK1-2, *Figure 1A*). Posterior probabilities of these reconstructions are shown in *Figure 1—figure supplement 2*. The full reconstruction files for all ancestors can be accessed at http://phylobot.com/350478581/msaprobs.PROTCATLG/ancestors.

### Evolution of substrate specificity in the MAP kinase lineage

We synthesized the coding sequences for these inferred ancestors and purified them from *E. coli*. We then used a positional-scanning peptide library (PSPL; *Hutti et al., 2004*) to determine the specificity of all inferred ancestors. The specificities of all inferred ancestors were consistent with expectations. As previously reported, the inferred deep ancestor of the CMGC group contained hallmarks of specificity found along multiple branches of the CMGC phylogenetic tree including +1 proline, −2 proline and −3 arginine preference. The new inferred CDK and MAPK ancestors generated for the current study lost the −3 arginine specificity at the transition to the inferred ancestor of CDK and MAPK (AncCDK.MAPK, *Figure 1—figure supplement 3A and B*). This was most likely due to the mutation of Glu to Ser in the conserved HRDxKPEN motif, previously reported to be a strong determinant of −3 arginine preference (*Mok et al., 2010*). All inferred ancestors had a strong preference for proline at the +1 position including the inferred ancestor of all MAP kinases (AncMAPK, *Figure 1B*), as expected. Together the specificities of inferred ancestors were concordant with expectations providing confidence in our reconstructed kinases.

### Regulatory evolution in the MAP kinase lineage

Next, we tested the basal activity of the reconstructed kinases using Myelin Basic Protein (MBP) as a generic substrate. We found that all inferred ancestors prior to and including AncERK1-5 had relatively high activity, while AncERK1-2 had low activity, almost comparable to modern ERK1/2 (*Figure 1C*, *Figure 1—figure supplement 4*). One alternative possibility for this decreased phosphorylation of MBP is that kinase specificity changed such that AncERK1-2 does not efficiently recognize the MBP substrate. However, PSPL specificity arrays indicated that there was very little change in specificity from AncERK1-5 to modern ERK2 (*Figure 1—figure supplement 3*). We also assessed the activity of AncERK1-5 and AncERK1-2 to a second substrate, ELK1. AncERK1-2 also had had much lower activity towards EL-1 than AncERK1-5 (*Figure 1—figure supplement 5A and B*), strongly indicating a change in kinase activity, not specificity. Thus, a transition in kinase activity occurred between AncERK1-5 and AncERK1-2. Importantly, AncERK1-2 was activated ~6 fold either by coincubation with MEK (*Figure 1C*) or by purification from an *E. coli* strain that coexpressed MEK (*Figure 1—figure supplement 5C and D*), indicating that the decrease in activity was a regulatory evolution event rather than a general problem with kinase reconstruction leading to loss of activity for trivial reasons such as failure to fold correctly. We believe that the modest activation of AncERK1-2 by MEK is due to the high specificity this activating kinase; AncERK1-2 is likely a poor MEK substrate compared to modern ERKs 1 and 2. The decrease in activity of AncERK1-2 was due to both an increase in the Michaelis-Menten constant ($K_M$) and a decrease in $k_{cat}$ (*Table 1*, *Figure 1—*

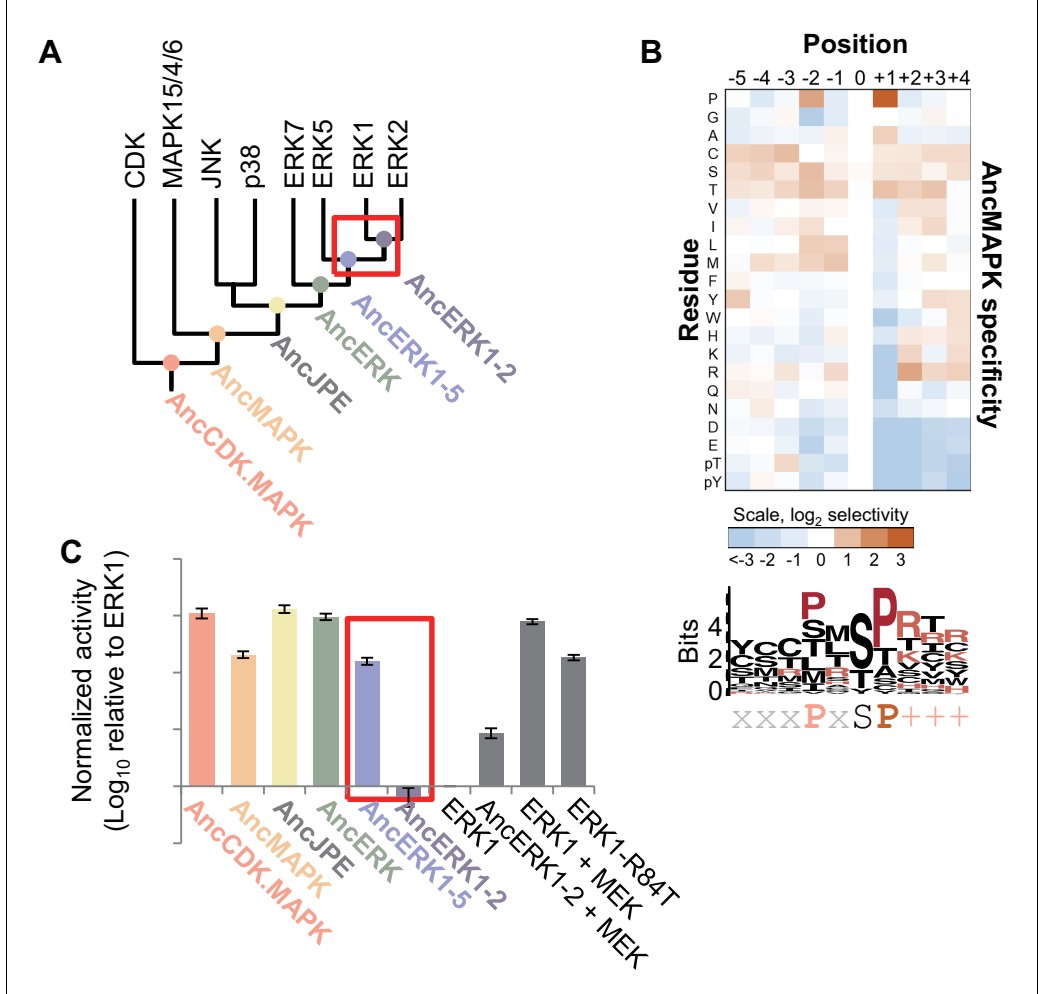

**Figure 1.** Kinase activity became more tightly regulated at the transition from AncERK1-5 to AncERK1-2. (**A**) Phylogenetic tree for the CDK. MAPK group. The ancestral nodes corresponding to the kinases reconstructed in this study are indicated by circles. The red box indicates a transition from high basal activity to low activity in the absence of activation by MEK. (**B**) Substrate specificity of AncMAPK. Average of two replicates positional scanning peptide libraries were used to profile the specificity of the inferred ancestral kinase AncMAPK. Red indicates positive selection, blue indicates negative selection. (**C**) Quantification of the basal activities of inferred ancestral kinases and modern ERK1 with and without the activating kinase, MEK ($log_{10}$ rate of myelin basic protein phosphorylation activity relative to ERK1), red box indicates the transition in basal activity. Error bars indicate ± SD, n = 3.

The online version of this article includes the following figure supplement(s) for figure 1:

**Figure supplement 1.** Summary of posterior probabilities of residues for AncCDK.MAPK, AncMAPK, AncJPE and AncERK.

**Figure supplement 2.** Summary of posterior probabilities of residues for AncERK1-2 and AncERK1-5.

**Figure supplement 3.** Determination of the primary sequence specificity of inferred ancestral kinases.

**Figure supplement 4.** Example primary data for kinase assays.

**Figure supplement 5.** The pattern of activities of AncERK-2 and AncERK1-5 towards a second substrate, ELK1, and activation of AncERK1-2 by MEK.

**Figure supplement 6.** Michaelis–Menten plots.

*figure supplement 6*). Therefore, we have identified a major transition in kinase regulation within the ERK kinase family.

**Table 1.** Michaelis-Menten kinetic parameters for AncERK1-5, AncERK1-2 and ERK1 with Myelin Basic Protein as a substrate.

| Kinase | $K_M$ ($\mu$M) | 95% confidence interval | $k_{cat}$ (min$^{-1}$) | 95% confidence interval |
|---|---|---|---|---|
| AncERK1-5 | 1.47 | 1.08, 2.00 | 0.64 | 0.59 to 0.69 |
| AncERK1-2 | 40.43 | 32.44, 50.63 | 0.0011 | 0.0010 to 0.0012 |
| ERK1 | 9.48 | 6.35, 13.98 | 0.0065 | 0.0059 to 0.0072 |

## The higher basal activity of deep inferred ancestors is due to autophosphorylation of the TxY motif

We next aimed to determine the mechanism of the activity transition between AncERK1-5 and AncERK1-2. Almost all MAPKs have a conserved 'TxY' motif at the activation lip in their kinase domain, and dual phosphorylation of this motif drives kinase activation. Dual phosphorylation of the threonine and tyrosine residues of 'TxY' motif in ERK2 is catalyzed by MEK1/2. This dual phosphorylation causes the N- and C-terminal domains in ERK2 to rotate toward one another, thus organizing the catalytic sites (*Canagarajah et al., 1997*). This conformational change also promotes substrate recognition in the p+1 site (*Figure 2A*).

Multiple sequence alignment analysis showed that the 'TxY' motif was conserved among all inferred kinase ancestors (*Figure 2—figure supplement 1*). To test the role of the 'TxY' motif in the evolution of kinase activity, we first mutated the 'TxY' motif to 'AxF', thus removing the phosphoacceptor hydroxyl groups, rendering the motif non-phosphorylatable. These mutations greatly reduced the activity of all inferred kinase ancestors (*Figure 2B*), suggesting that requirement for 'TxY' phosphorylation was an early evolutionary event. Next, we quantified the phosphorylation level of threonine and tyrosine in the 'TxY' motif in reconstructed ancestral kinases and ERK1 by liquid chromatography mass spectrometry of tryptic peptides (LC-MS, *Figure 2C*). Consistent with the kinase activity transition, we observed that the level of phosphorylated tyrosine (Tx-pY) was relatively high in deeper inferred ancestors: AncCDK.MAPK (22.9%), AncMAPK (32.3%), AncJPE (48.1%), AncERK (76.3%) and AncERK1-5 (76.0%), and then dropped precipitously at the transition to AncERK1-2 (5.7%), which had levels similar to modern Erk1 (3.8%). Phosphorylation of threonine in the 'TxY' motif (pT-xY), followed a similar pattern of higher levels in inferred deep ancestor kinases, including AncCDK.MAPK (35.1%), AncMAPK (7.6%), AncJPE (12.4%) and AncERK (7.3%), and almost undetectable levels in AncERK1-5, AncERK1-2 and modern Erk1. Notably, we also detected di-phosphorylation of the 'TxY' motif (pT-x-pY) in the inferred deep ancestor kinases AncCDK.MAPK (6.3%), AncJPE (27.6%) and AncERK (7.9%). These results suggest that autophosphorylation at the 'TxY' loop became much less efficient in AncERK1-2, thus leading to dependence on upstream activating kinases.

## Shortening of linker loop and mutation of gatekeeper residue led to tight regulation in the ERK1-2 family

To understand how this regulatory evolution occurred, we next sought to identify the specific mutations underlying the transition in kinase activity that occurred between AncERK1-5 and AncERK1-2. First, however, we wished to determine whether the activities of AncERK1-5 and AncERK1-2 were robust to uncertainties in ancestral sequence reconstruction. To address this issue, we synthesized alternative inferred ancestors combining all alternate residues with posterior probability >0.2 (*McKeown et al., 2014*). The inferred ancestors of alternative kinases behaved similarly to the original inferred ancestors (*Figure 3—figure supplement 1*). The alternative ancestor Alt.AncERK1-2 had 2.5-fold higher activity than the maximum likelihood AncERK1-2, possibly due to slight differences in the activation segment and the L16 loop. Nevertheless, Alt.AncERK1-2 still has 100-fold lower activity than AncERK1-5 and both of its alternative reconstructions. These results further suggest that the transition of kinase activity occurred between AncERK1-5 and AncERK1-2 and that our results are robust to phylogenetic and reconstruction uncertainty (*Hanson-Smith et al., 2010*).

With this test, we were confident to compare inferred ancestors in an attempt to identify causal evolutionary mutations. As an initial strategy, we constructed a series of chimeras between AncERK1-5 and AncERK1-2. First, we divided the inferred ancestor kinases into three parts (N

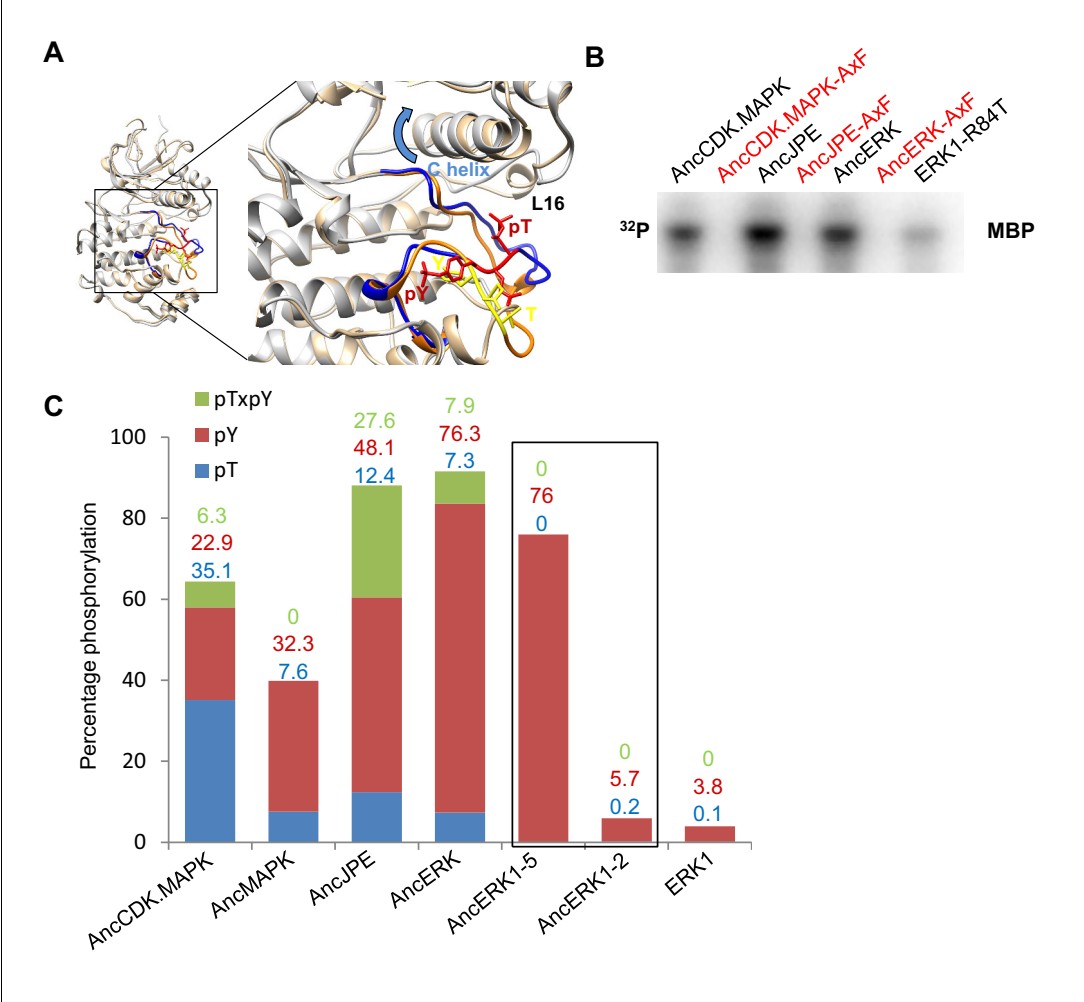

**Figure 2.** Higher basal activity of inferred ancestors is due to autophosphorylation on the TxY motif. (**A**) Superposition of ERK2 (gray, Protein Data Bank code 1ERK) and phosphorylated ERK2 (tan, Protein Data Bank code 2ERK). The activation segment (ribbon diagram) and the 'TEY' motif (represented as sticks) are colored orange and yellow for ERK2, blue and red for phosphorylated ERK2. Phosphorylation of the TEY motif leads to a conformational change that remodels the activation loop to enable substrate binding, and rotation of the C helix (arrow) leading to interaction between Lys (K52) on the β3 strand and Glu (E69) on the C helix. Part of Loop L16 adopts a helical conformation in phosphorylated ERK2. (**B**) Kinase activity of wildtype and AxF mutants of inferred ancestors towards MBP. (**C**) LC-MS quantification of phosphorylation levels threonine and tyrosine in the 'TEY' motif in reconstructed ancestral kinases and ERK1. Bar graphs indicate the relative abundance of various phosphorylated forms of tryptic peptides that contain the 'TEY' motif.

The online version of this article includes the following figure supplement(s) for figure 2:

**Figure supplement 1.** Sequence alignment of Reconstructed ERK1-2 kinase ancestors and modern ERK1.

terminus, middle, and C terminus, *Figure 3—figure supplement 2A and B*), and shuffled these parts (*Figure 3—figure supplement 2B*, chimera 1–3). One strong possibility was that changes in the flexibility of the activation loop would account for differential TxY autophosphorylation. However, chimeras that swapped the middle domain of the inferred ancestors had no effect on activity. Rather, our initial chimeras indicated that changes in the N-terminus were responsible for most of the difference in the activity of these inferred ancestors (*Figure 3—figure supplement 2D*). We narrowed the causal variation using two additional chimeras (*Figure 3—figure supplement 2B and D*, chimera 4 and 5). A key advantage of ancestral reconstruction is that the number of amino acid transitions between inferred ancestors is typically far lower than the number of differences between extant proteins (*Thornton, 2004*). Therefore, after our chimera analysis there were only two possible changes to assess within the causal fragments, an insertion of an arginine after the 44th amino acid (ins44R,

AncERK1-2 numbering) and a transition from glutamine to methionine at the 89th amino acid (Q89M, AncERK1-2 numbering). Indeed, reverting these amino acids to the inferred ancestral state in AncERK1-2 (referred to as the AncERK1-2-ins44R-Q89M mutant) was sufficient to confer high basal activity to this kinase (*Figure 3A*). This increased activity is almost as high as ERK1 activated by MEK, or the activated ERK1 mutant suggesting that reversion of these two evolutionary changes to the inferred ancestral state makes AncERK1-2 mostly independent of activating kinases. Adding MEK to the AncERK1-2-ins44R-Q89M does not increase the kinase activity much. Adding MEK to AncERK1-2 increases activity ~6 fold, but this leads to < 10% of the activity of the active mutant. We believe that this inefficiency is due to the known high specificity of MEK for ERK1 and 2. We speculate that MEK doesn not recognize AncERK1-2 as efficiently as ERK1 and 2, accounting for the lower efficiency of activation of AncERK1-2. Conversely, introducing just these two mutations into AncERK1-5 (i.e. deletion of R46 and mutation of M90 to Q) suppressed basal activity to the levels of AncERK1-2 (*Figure 3—figure supplement 2C and E*). By modeling the AncERK1-2 sequence onto the Erk2 structure (PDB: 1ERK), we mapped the 44R insertion to the loop between the β3 strand and helix αC, and the Q89M mutation to the gatekeeper position at the back of the ATP-binding

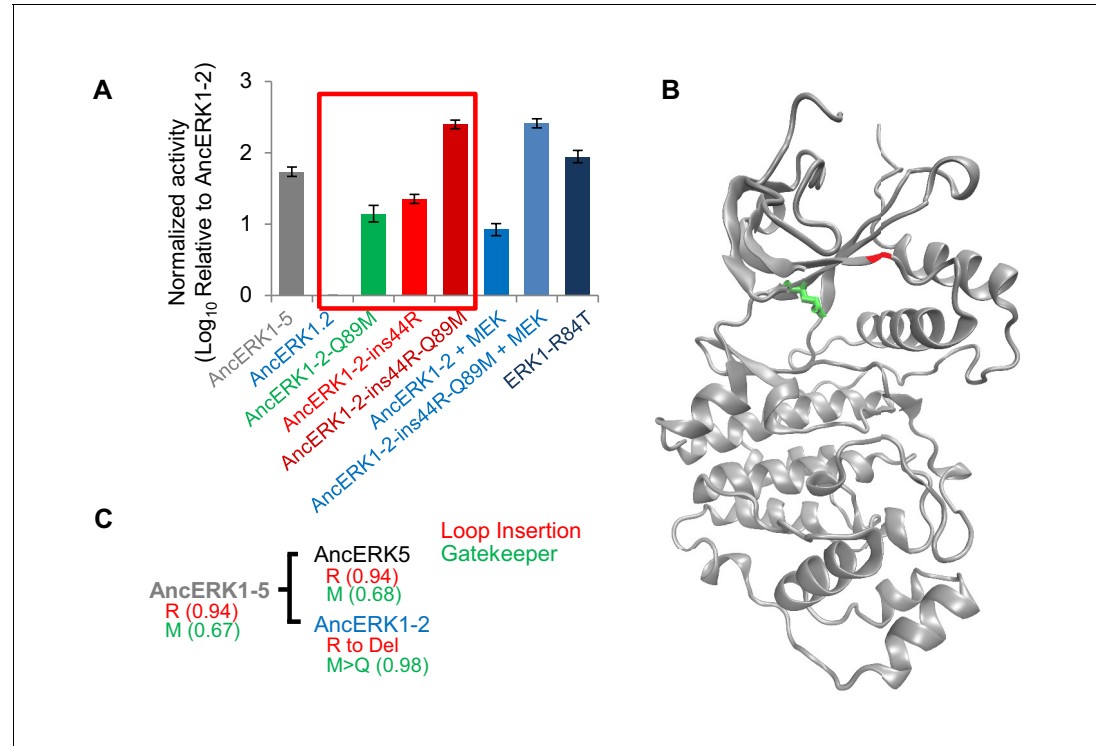

**Figure 3.** Shortening of the linker loop and mutation of the gatekeeper residue led to tight regulation in the ERK1-2 family. (**A**) Kinase assays comparing the activity of two evolutionary mutants to the activity of AncERK1-2 before and after preincubation with active MEK kinase (log$_{10}$ rate MBP phosphorylation). Error bars indicate ± SD, n = 3. (**B**) Positions of the two evolutionary mutations modeled on the ERK2 structure (PDB 1ERK). The gatekeeper residue, Q122, is shown in green, and the location of the 74N insertion is shown in red. (**C**) Phylogenetic tree indicating the state of the β3-αC insertion and identity of gatekeeper residues during evolution of the AncERK1-5 group. Numbers in the brackets indicate the posterior probability for ancestral reconstructions.

The online version of this article includes the following figure supplement(s) for figure 3:

**Figure supplement 1.** Activities of inferred ancestors are robust to uncertainties in reconstruction.

**Figure supplement 2.** Identification of the evolutionary mutations mediating the decrease in basal activity between AncERK1-5 and AncERK1-2.

**Figure supplement 3.** Alignments showing the position of the mutations that drove the transition in activity that occurred between AncERK1-5 and AncERK1-2.

**Figure supplement 4.** Phylogenetic tree indicating the mutations that occurred during the evolution from the inferred common ancestor of CDK and MAPK to modern kinases.

**Figure supplement 5.** Slight activation of wildtype AncERK1-5 and AncERK1-5 double mutant by MEK.

pocket (*Figure 3B*). Together, these results suggested that Q89M and ins44R were the key mutations in the evolutionary transition from high to low basal activity.

## Reversing evolutionary mutations leads to activation of modern ERK1

The β3-αC loop and gatekeeper mutations are innovations of the ERK1-2 subfamily. All the inferred ancestors prior to AncERK1-2 had a longer linker loop and a hydrophobic residue at the gatekeeper position (*Figure 3C*; *Figure 3—figure supplements 3* and *4*). Therefore, we wondered whether mutating these residues back to an inferred ancestral state might be sufficient to increase modern ERK1/2 activity. First, we introduced the insertion into the loop between the β3 strand and αC helix in ERK1. The insertion increased ERK1 activity more than 30-fold (*Figure 4A*; *Figure 4—figure supplement 1*). We also found that the Q122M gatekeeper mutant led to ERK1 activation (*Figure 4A*) and phosphorylation of the 'TEY' motif was enhanced (*Figure 4B*). Mutation in the gatekeeper residue has been reported to activate ERK2 in previous studies (*Emrick et al., 2006*), but this mutation was to ALA or GLY, therefore it is interesting that both small residues and the evolutionary mutation to a bulky polar residue activate ERK1. When we combined the loop insertion and Q122M gatekeeper mutation, the double mutant was much more active than either single mutant (*Figure 4A*). Indeed, the combination of these two mutations raised ERK1 activity by more than 2 orders of magnitude, almost to the levels of ERK1 after activation with MEK. Therefore, reversing just these two evolutionary changes in modern ERK1 reverts the kinase back to an inferred ancestral regulatory state with very little dependence on its upstream regulatory kinase.

## The length of the β3-αC loop determines basal activity

We wondered if the identity of the amino acid insertion after position 44 of ERK1 was important. Our reconstruction showed that arginine was not the only possible amino acid insertion at position 44. Asparagine was the next most likely alternative inferred ancestral amino acid at this position. Therefore, we tested an ERK1-74N insertion mutant (ins74N in ERK1 numbering) and determined this mutant to be just as active as the ERK1-74R allele. Indeed, we found that, in addition to arginine and asparagine, many other amino acid insertions also increased ERK1 activity (*Figure 4—figure supplement 1*). Thus, we hypothesized that loop-length rather than precise amino acid identity might explain the increased kinase activity. To test this hypothesis, we inserted and deleted residues along this loop. Deletions and insertions from K72 to P75 (ERK1 numbering) greatly affected mutant activity. However, length changes after P75, had a minor effect (*Figure 4—figure supplement 2A and B*). These results confirmed the importance of loop length from K72 to P75 in the determination of ERK1 basal activity.

## Increased activity in modern ERK1 is due to *cis* autophosphorylation on the TxY motif

Because the 'TxY' motif was highly phosphorylated in inferred deep ancestral kinases, we hypothesized that increased 'TxY' phosphorylation would likely also be a mechanism of mutant ERK1 activation. Consistent with this idea, mutations in this motif to 'AxY', 'TxF' and 'AxF' all abolished the increased activity caused by the insertion mutation (*Figure 4C*). Furthermore, Western blots using an antibody specific for phosphorylated ERK (*Figure 4B*) and LC-MS quantification of tryptic peptides (*Figure 4D*) both showed elevated 'TxY' phosphorylation in mutant kinases.

The *E. coli* bacterium that we use to express ERK1 and inferred ancestral kinases lacks endogenous kinases capable of phosphorylating the 'TxY' motif. Therefore, the increased phosphorylation of 'TEY' motif is likely caused by autophosphorylation (*Howard et al., 2014*). Indeed, experiments incubating kinase with ATP-y-$^{32}$P in the absence of substrate showed that autophosphorylation rates were indeed increased in the ERK1-ins74N mutant (*Figure 4—figure supplement 3*). Together, these results suggested that the insertion caused elevated phosphorylation of the 'TxY' motif. To test whether this autophosphorylation occurred in *cis* or in *trans*, we employed a catalytically inactivated mutant, ERK1-ins74N-K52R. We fused the kinase dead allele to GST to increase the molecular weight of this protein allowing us to distinguish it from catalytically active ERK1 on an SDS-PAGE gel. We then mixed the kinase dead allele with the activated ERK1-ins74N allele. There was no detectable phosphorylation of the kinase-dead GST-ERK1-ins74N-K52R, even though the catalytically active allele clearly continued to undergo autophosphorylation (*Figure 4E*). To provide further

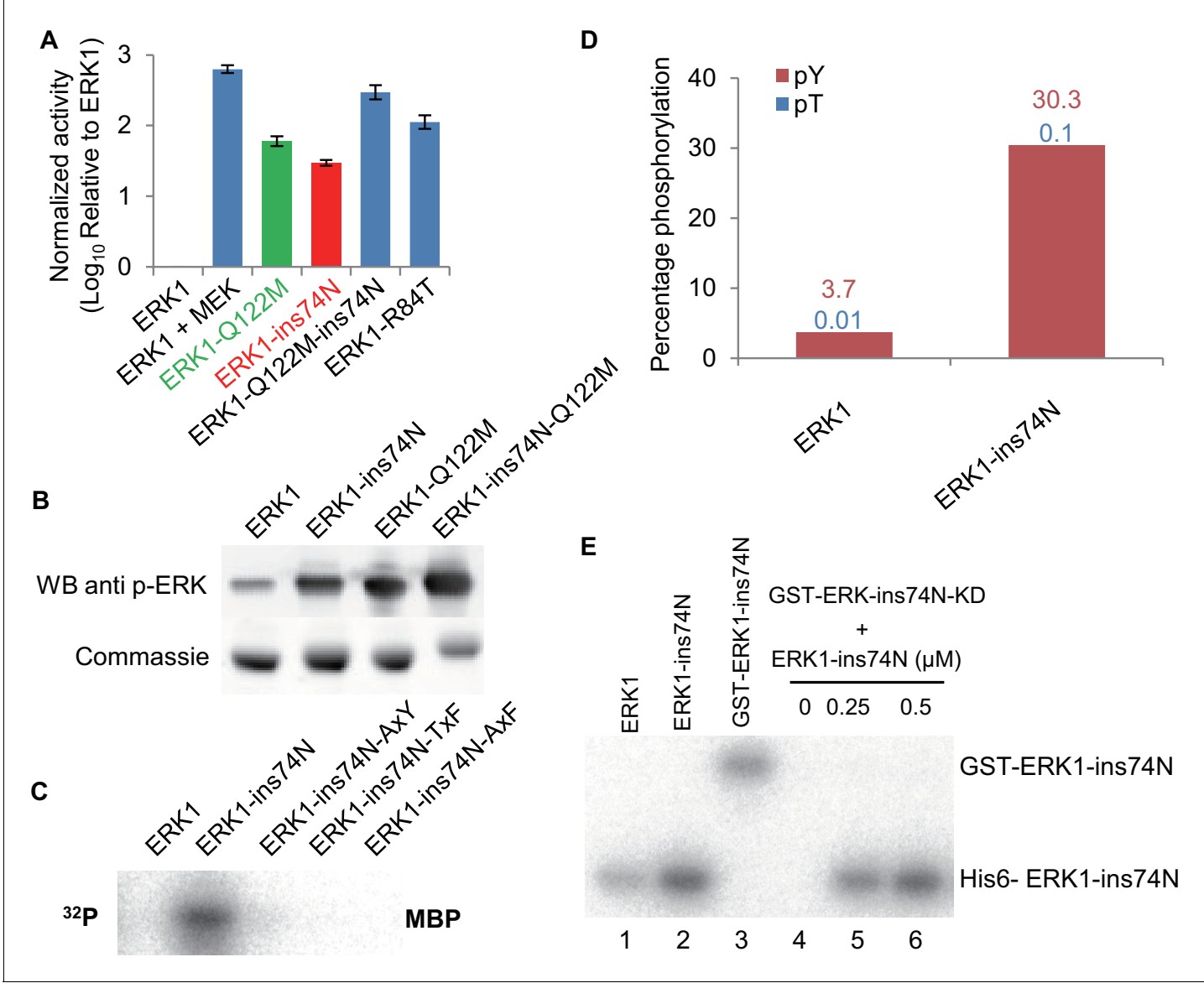

**Figure 4.** Reversing evolutionary mutations leads to activation of modern ERK1 through *cis*-autophosphorylation of the TEY motif. (**A**) Kinase assays measuring activity of modern ERK1 and mutants that revert the β3-αC loop length and gatekeeper residue to the inferred ancestral state (log$_{10}$ rate of MBP phosphorylation relative to ERK1). Error bars indicate ± SD, n = 3. (**B**) Western blots indicating degree of phosphorylation of the residues in the kinase activation loop (top), Coomassie brilliant blue loading control (bottom). (**C**) Activity of ERK-ins74N mutant and its 'TEY' mutants towards MBP. (**D**) Quantification of the phosphorylation level of the 'TEY' motif in ERK1 wildtype and ins74N mutant by LC/MS analysis of tryptic peptides. (**E**) Autophosphorylation assay. 0.5 µM Erk kinase and its mutants were incubated with kinase assay buffer without substrate. The ERK1 kinase and the ins74N mutant were tagged with either His6 (lanes 1 and 2) or GST (lane 3, the GST tag was used to alter the mobility of kinase alleles allowing them to be distinguished on an SDS-PAGE gel). GST-ERK1-ins74N kinase dead K71R mutant (ins74N KD) was incubated without His6-ERK1-ins74N (lane 4), and with 0.25 µM or 0.5 µM His6-ERK1-ins74N (lanes 5 and 6).

The online version of this article includes the following figure supplement(s) for figure 4:

**Figure supplement 1.** Other amino acid insertions after ERK1 residue 74 also increase basal activity.
**Figure supplement 2.** The length of the linker loop between K71 and P75 determines kinase activity.
**Figure supplement 3.** The ERK1 ins74N mutant has increased autophosphorylation on the TEY motif.
**Figure supplement 4.** Autophosphorylation rate of ERK1-ins74N scales linearly with kinase concentration.

support for an intramolecular mechanism, we measured autophosphorylation rates as a function of ERK1 concentration. *Cis*-autophosphorylation should scale linearly with ERK1 concentration, while *trans*-autophosphorylation should scale as the square of ERK1 concentration. We found the former result to be true (*Figure 4—figure supplement 4*). These results imply that autophosphorylation occurs through an intramolecular mechanism in *cis*.

## Molecular dynamics simulations reveal increased flexibility at the glycine rich loop and loop 16, and a higher probability of an active organization of catalytic residues in mutant kinases.

To investigate if changes in protein dynamics might suggest mechanisms by which autophosphorylation was increased, we performed a series of ensemble molecular dynamics (MD) simulations. We focused on the unphosphorylated structure, which is only available for ERK2 (PDB ID:1ERK; *Canagarajah et al., 1997*). First, we confirmed that the evolutionary mutations identified in ERK1 also led to activation of ERK2 (*Figure 5—figure supplement 1*). Indeed, the ins55N and gatekeeper also activate ERK2, further supporting our evolutionary hypothesis. We then performed MD simulations of wild-type ERK2, the ß3-αC loop insertion mutant (ins55N, rat ERK2 numbering) in ERK2, the gatekeeper residue mutation Q103M (rat ERK2 numbering), and the double mutant (ins55N + Q103M).

We performed 1000 independent MD simulations on Folding@home (*Shirts and Pande, 2000*) for ERK2- WT, ins55N, Q103M and the ins55N-Q103M double mutant starting from the unphosphorylated ERK2 structure (PDBID: 1ERK), which is in the inactive, 'DFG-in' conformation. These 4,000 MD simulations were propagated for a total of at least 500 µs per system, resulting in an aggregate total of ~2.05 ms of simulation time (*Figure 5—figure supplement 2*). For each system, we assessed multiple hypotheses for how mutations might affect *cis*-autophosphorylation. Potential mechanisms included: changes in the overall flexibility of the kinase fold, decreases in the relative distances between the key catalytic residues in the kinase active site, reorganization of the 'C-spine' and 'R-spine' (*Kornev et al., 2006*; *Taylor and Kornev, 2011*), and a reduced domain closure angle between the N and C-terminal domains (*Kornev et al., 2006*; *Meharena et al., 2013*) (structural features are illustrated in *Figure 5A*). In addition, we assessed a previously suggested mechanistic pathway (*Emrick et al., 2006*) wherein mutations in the gatekeeper residue affect a cluster of hydrophobic interactions that change the flexibility of the 'activation lip', the part of the activation segment that contains the TEY phosphorylation motif.

First, we compared the overall flexibility (indicated by the αC root mean square fluctuations, RMSF) of each mutant to WT. ERK2-ins55N showed increased flexibility at the glycine rich loop (*Figure 5B*), ERK2-Q103M showed increased flexibility at loop 16 (L16, *Figure 5C*) and the ERK2-ins55N-Q103M double mutant was more flexible at both the glycine rich loop and L16 (*Figure 5D*). Therefore, each mutation has subtle effects on peptide backbone dynamics and the double mutant combines both of these effects.

We next investigated the dynamics of the catalytic residues. There are several residues critical for catalysis: K52/71, D147/166 and D165/184 (rat ERK2/human ERK1 numbering) (*Roskoski, 2012*). The salt bridge between the ß3-Lys52 and αC-Glu69 (K52-E69) promotes kinase activity and corresponds to the 'αC-helix in' active conformation (*Roskoski, 2012*), thus the distances between these residues are informative for the expected catalytic rate of the active site. Indeed, the K52-E69 salt bridge was more frequently present in mutants than WT (*Figure 5E*). Next, we studied other inter-residue distances in the catalytic site, K52-D165 and E69-F166 from the DFG motif. Here, average inter-residue distances were significantly reduced in at least one of the mutants (*Figure 5—figure supplement 3A and B*). The ERK2-ins55N and ERK2-Q103M mutants both altered these average distances, and the ERK2-ins55N-Q103M double mutant was roughly the sum of the effects consistent with the additive effect of mutations observed in our experiments. This result is consistent with the idea that the catalytic site of ERK2 can more readily adopt an active conformation when reverted to the inferred ancestral state, thereby increasing the rate of autophosphorylation.

The gatekeeper residue lies between two conserved non-consecutive structural 'spines' of hydrophobic amino acids, the R- and C-spines that connect the two lobes and reorganize upon kinase activation (*Taylor and Kornev, 2011*). We therefore analyzed the organization of the R-spine and C-spine. We did not observe statistically significant differences in distances along the R-spine residues between the WT and mutant simulations. However, there were some changes in C-spine

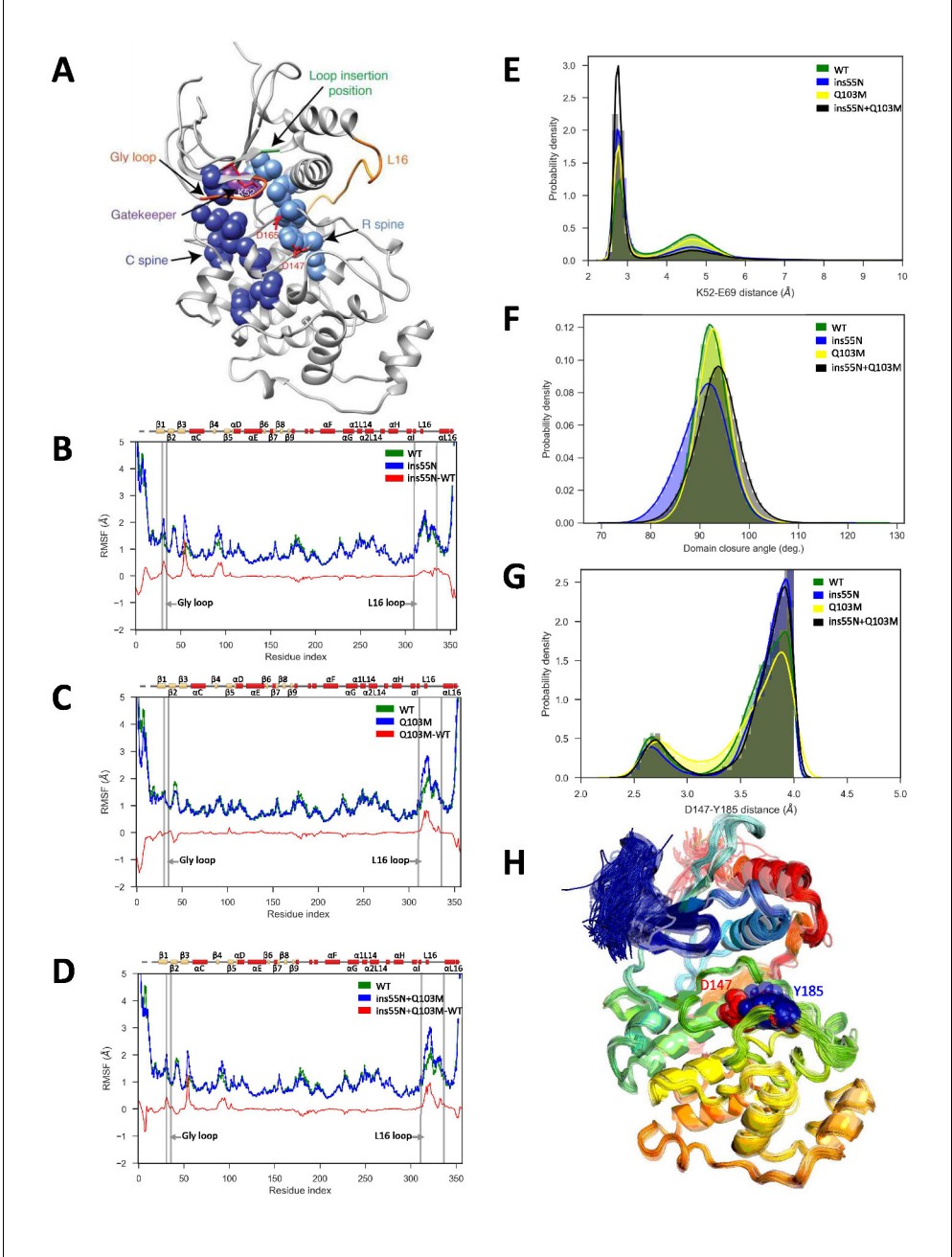

**Figure 5.** Molecular dynamics simulations of activated mutants show increased flexibility at the glycine rich loop and loop 16, and increased salt-bridge formation at the catalytic site. (A) Structure of ERK2 with relevant features annotated: L16 – orange, Gly loop - dark orange, C spine - dark blue, R spine - light blue, gatekeeper residue – purple, catalytic residues (K52, D147 and D165) – red, and loop insertion position – green. (B–D) Root mean square fluctuations (RMSF, error bars indicating 95% confidence intervals of the means are plotted but are too small to see): Wild type – green, mutant – blue and difference between mutant and WT – red. (B) ERK2-ins55N (ß3-αC loop insertion mutant) (C) ERK2-Q103M (gatekeeper residue mutant) (D) ERK2-ins55N-Q103M (double mutant). (E) Probability density of salt-bridge between ß3-lysine and αC-glutamate, K52–E69. (F) Distribution of domain closure angles. (G) Probability density of distances < 4 Å between Y185 of the 'TxY' motif and active site D147H: Wild type – green, ERK2-ins55N – blue, ERK2-Q103M – yellow and ERK2-ins55N-Q103M – black. (H). Top 100 conformations of ERK2-WT with smallest Y185-D147 distance. D147 (red) and Y185 (blue) are shown as spheres.

The online version of this article includes the following figure supplement(s) for figure 5:

*Figure 5 continued on next page*

*Figure 5 continued*

**Figure supplement 1.** Reversion of evolutionary mutations also activates ERK2.

**Figure supplement 2.** Distribution of trajectory lengths.

**Figure supplement 3.** Probability densities of various inter-residue distances in the catalytic site.

**Figure supplement 4.** Differential contacts between ERK2 mutants and wild type (WT).

**Figure supplement 5.** A 3-State Markov state model (MSM) indicates high contact disruption along the activation segment in the ERK2-Q103M mutant.

**Figure supplement 6.** A 4-State Markov state model (MSM) indicates an increase in the open conformation of loop 16 in insertion mutants.

**Figure supplement 7.** Major contact changes with Y185 mapped on to the structure of mutants.

---

distances. Distributions of these distances are plotted in *Figure 5—figure supplement 3C–E*. Gate-keeper mutations were previously shown to activate kinases through changes in the R-spine conformation (*Azam et al., 2008*). However, these studies focused on activation mutations that lead to drug resistance, which are from smaller to larger gatekeeper amino acids (*Tamborini et al., 2004*; *Azam et al., 2008*), while our mutation is between similarly sized but chemically distinct amino acids (glutamine to methionine). Therefore, this mechanism may not be at play. It is also quite possible that our MD simulations are too short to capture changes in the R-spine.

Previous MD simulations have shown changes in angles between the N- and C-terminal domains in gatekeeper mutants of ERK2. This domain closure is a key step in kinase activation (*Barr et al., 2011*). Therefore, we studied domain closure by calculating the angle between the C and E helices (*Figure 5F*). We found a wider distribution of angles in the ERK2-ins55N and ERK2-ins55N-Q103M mutants compared to the WT, suggesting an increase in inter-domain flexibility. However, a true understanding of the relevance of our mutations to domain closure will require much longer simulations as NMR measurements have shown that these motions occur on the millisecond timescale (*Xiao et al., 2014*).

We next took an unbiased approach to identify contacts that were gained or lost by determining a differential contact map between the mutant and wild type kinases for all residue pairs (*Figure 5—figure supplement 4*). In all three mutant simulations, we observed contact changes compared to WT along the activation lip that includes the phosphorylation sites (T183 and Y185). Furthermore, in ERK2-Q103M and ERK2-ins55N-Q103M double mutant simulations, we also observed changes in contacts between the L16 loop (Res.310–336) and αE (Res. 121–141) regions. Additionally, in simulations with the ß3-αC loop insertion mutant (ERK2-ins55N) and the ERK2-double mutant (ERK2-ins55N-Q103M) we observed contact changes between the ß3-αC loop and glycine-rich loop regions (Res. 30–35). We visualized residues that undergo large contact changes on the ERK2 structure (*Figure 5—figure supplement 4D–F*). This analysis indicated that these changes are mostly in the activation lip and in regions close to mutation sites in the N-lobe. These results are consistent with the changes in RMSF observed above, and additionally implicate changes in activation loop dynamics as a potential mechanism for increased autoactivation.

To gain a deeper understanding of the dynamical impacts of mutations, we built Markov state models (MSMs) from a set of 125 highly informative inter-residue distances (see Materials and methods). MSMs help integrate information from multiple simulations to reveal the system's kinetically distinct clusters of conformations, which are referred to as macrostates (*Chodera et al., 2007*; *Pande, 2010*). We used MSMs to determine which macrostate populations can be perturbed by mutations. We first built MSMs for ERK2-WT and Q103M systems (*Figure 5—figure supplement 5*) and identified three macrostates. We found that the Q103M mutation increased the populations of macrostates A and C, which had high contact disruption along the activation segment. Macrostate A had the highest overall contact disruption among the three states, and to a lesser extent macrostate C showed a pattern of changes in the activation segment similar to macrostate A, but also had increased disruptions along the L16 loop. We then identified four-state MSMs for the ß3-αC loop insertion and double mutant systems (ERK2-ins55N and ERK2-ins55N + Q103M) (*Figure 5—figure supplement 6*). Comparing the double mutant MSM to ins55N, we found an increase in the population of macrostate D in the ins55N mutant, which had very little change along the L16 loop but featured an alternative, 'open', conformation of the ß3-αC loop. These MSMs present a vivid picture of the structural changes caused by evolutionary mutations and

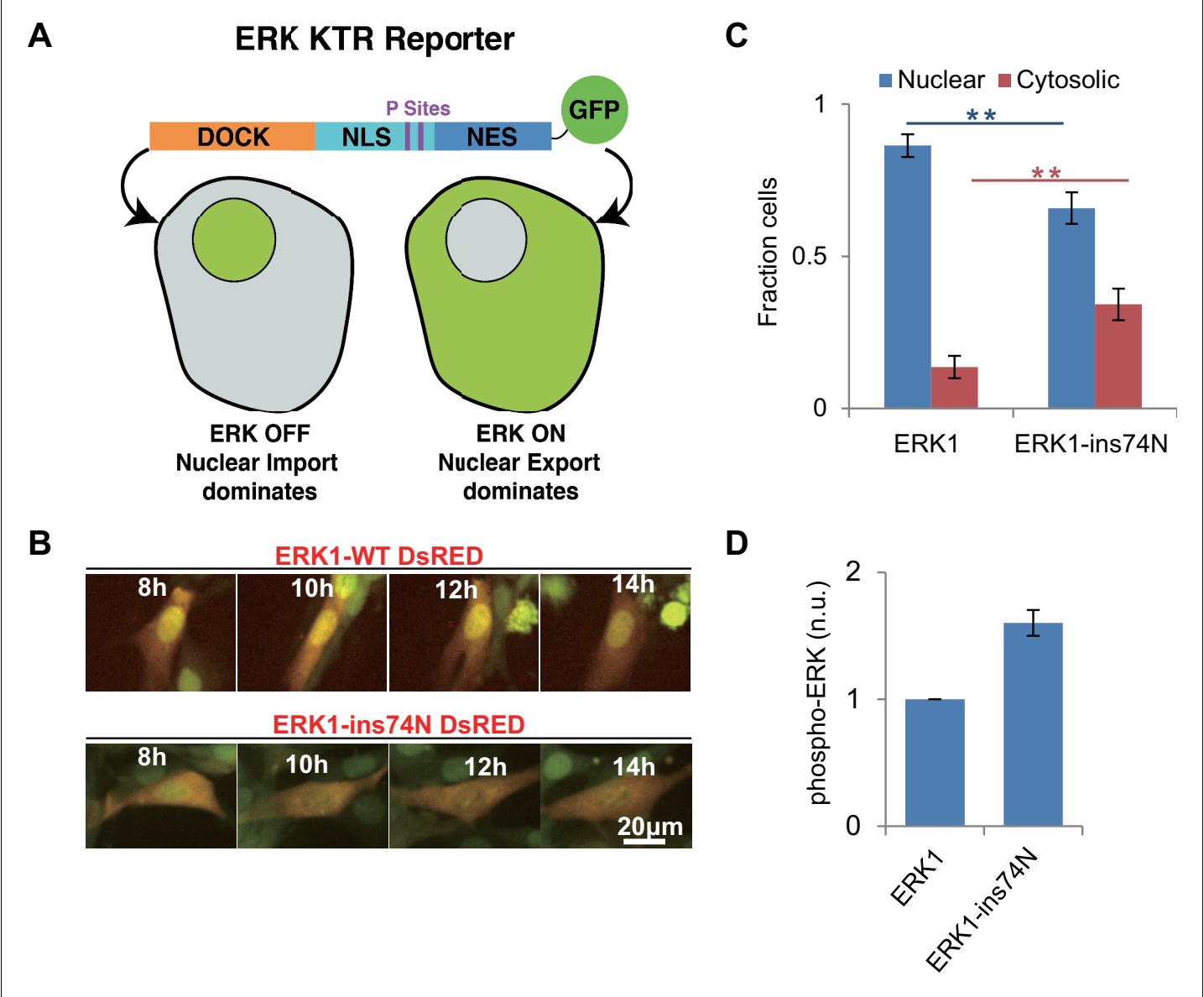

**Figure 6.** Reversion of ERK regulatory mutations to the inferred ancestral state leads to MEK independent activity in vivo. (**A**) Schematic of the Kinase Translocation Reporter (KTR): phosphorylation of the reporter by ERK1/2 leads to inactivation of a nuclear localization signal (NLS) and activation of a nuclear export signal (NES), leading to nuclear export. Conversely, in cells with low ERK1/2 activity, the NLS is active and NES inactive, therefore the reporter is localized to the nucleus. (**B**) 3T3 cells expressing ERK1 (indicated by coexpression of DsRED with an IRES, after the ERK1 ORF) and the ERK-KTR (green) were treated with 1 μM MEK inhibitor (Trametinib) and grown without FBS. (**C**) Quantification of ERK-KTR localization after 12–14 hr Trametinib treatment and FBS deprivation. The fraction of cells in which the majority of ERK-KTR localizes to the nucleus or cytosol is plotted. Statistical comparisons are by Student's t-test, **p<0.01. Error bars indicate ± SD, n = 3, biological replicates (40–50 cells per replicate). (**D**) Quantification of western blot to assess phosphorylation of the ERK TEY, normalized to the ppERK1 fraction in WT ERK1. Protein was extracted after 12–14 hr Trametinib treatment and FBS deprivation.

The online version of this article includes the following figure supplement(s) for figure 6:

**Figure supplement 1.** Raw data from *Figure 6D*.

further implicate increased flexibility in the activation segment as a potential mechanism for increased activity of mutants.

Finally, we compared our simulation data to a previously proposed mechanism (*Emrick et al., 2006*) where gatekeeper mutations were shown to control the accessibility of conformations able to autophosphorylate the activation lip. In all systems, we found a low population of conformations (up

to 1.7%) where the Y185 of the 'TxY' motif is hydrogen bonded to the active site D147, an interaction required for autophosphorylation. A large displacement of the Y185, which resides in a hydrophobic pocket, 8.3 Å away from D147 in the crystal structure, is required to achieve this contact. We found these conformations sampled in many of our trajectories (e.g. 325 out of the 1,000 WT trajectories). Unexpectedly, the frequency of these conformations is the highest in WT (1.7%) and decreases for the mutants (0.9% for 55N, 0.4% for Q103M, and 0.2% for 55N-Q103M) (*Figure 5G*, which shows only D147-Y185 distances of <4 Å). The top 100 conformations from the ERK2-WT system with the smallest D147-Y185 distance are shown in *Figure 5H*. While demonstrating the structural feasibility of autophosphorylation, these results don't implicate increased contact frequencies between D147 and Y185 as the mechanism for increased autophosphorylation in our case. We note that our simulations do not include ATP, which could affect these conformational equilibria. We examined additional contacts proposed (*Emrick et al., 2006*) to influence the stability of the activation lip: V186-L168, T188-D147 and T188-K149. Of these, only V186-L168 showed significant differences in the distributions of the distances in mutants compared to WT (*Figure 5—figure supplement 3G*). Consistent with the unexpected decreased frequency of conformations with the Y185 rearranged for hydrogen bonding with D147 in our mutants, the population of conformations with the V186-L168 contact perturbed is highest in WT (42%) and decreases for the mutants (24% for 55N, 29% for Q103M, 22% for 55N-Q103M). In light of these results and the earlier observation of contact changes along the activation lip compared to WT, we sought to examine those changes in more detail. We hypothesized that contact changes to Y185 would be a part of any true autophosphorylating conformations. We therefore specifically examined residues that have large changes in contact frequencies with Y185 (*Figure 5—figure supplement 7*). For all three mutants, the same seven residues had absolute mean contact changes larger than 0.05: L168, R192, I196, K201, G202, I207, and Y231. Of those, I207 and Y231 lost contacts with Y185 in all mutants, and R192 gained contacts in all mutants. We hypothesize that these residues might tune the rate of the autophosphorylation. It will be interesting to experimentally test these ideas in the future.

While these MD simulations were not long enough to observe the full transition to an active kinase conformation (*Gilburt et al., 2017*), they pave the way for future adaptive and enhanced sampling efforts, and offer hypotheses for further mutations to test experimentally. For now, our MD data primarily support the hypothesis that evolutionary mutations tune the contact frequencies of catalytic residues in the kinase active site. It is likely that additional mechanisms will be important at longer timescales, which warrant further investigations.

## Reversal of evolutionary regulatory mutations leads to MEK independent ERK activity in vivo

Next, we sought to determine whether the evolutionary insertion mutation led to higher ERK1 activity in vivo. We generated lentivirus encoding ERK1-WT or ERK1-ins74N and transduced NIH 3T3 cells with these two alleles. We used an ERK-KTR reporter (*Regot et al., 2014*), as our reporter for ERK kinase activity (*Figure 6A*). This reporter contains an ERK docking site and phosphoacceptor sites within nuclear localization and export sites. The reporter is designed such that it is localized in the nucleus when unphosphorylated and then translocates to the cytosol when phosphorylated by ERK. Before imaging, cells were pretreated with the MEK inhibitor Trametinib to prevent activation by endogenous mechanisms. When we performed time lapse imaging of cells to observe the localization of the ERK1-KTR reporter, we found that the ERK-KTR reporter was mainly localized in the nucleus in ERK1-WT cells, and only a small percentage of cells indicated any ERK activity (*Figure 6B and C*). This confirms that WT-ERK activity was low after inhibition of MEK by Trametinib. In contrast, cells expressing ERK1-ins74N had significantly more ERK-KTR in the cytoplasm. This result suggests that the evolutionary mutation led to MEK-independent ERK1 activity in vivo.

We also analyzed the phosphorylation level of the two ERK variants in vivo, by western blot using an antibody specific to phosphorylated ERK. NIH 3T3 cells transduced with both wildtype and mutant ERK were treated with Trametinib. After treatment, proteins were extracted for western blot. The western blot result showed that the phosphorylation level of 'TEY' motif in wild type ERK1 was very low after treatment with Trametinib, while the cells expressing ERK1-ins74N exhibited robust phosphorylation (*Figure 6D*, *Figure 6—figure supplement 1*). These results are consistent with the KTR reporter results, and together show that reversal of the evolutionary insertion mutation led to a decreased dependence on the upstream activating kinase MEK for ERK activity in vivo.

## Discussion

### Regulatory evolution in the MAP kinase group

Over the past several decades biochemical, genetic and structural studies have provided deep insights into the regulation of kinases. However, the mechanisms by which this diverse array of regulatory mechanisms evolved from a common ancestor remain poorly understood. In this study, we analyzed the regulatory evolution of the ERK kinases that lie within the CMGC group. Ancestral reconstruction revealed that the ability of these kinases to autophosphorylate at their activation loop sharply decreased at the split between the ERK5 and ERK1-2 kinase subfamilies. While the basal activity of the ERK5 subfamily remained relatively high, the ERK1-2 kinase subfamily became strongly dependent on the upstream activating kinase MEK. We identified two mutations that together account for most of the difference between modern ERK1-2 and ERK5. Strikingly, reversal of these evolutionary mutations in modern ERK1 led to high kinase activity and autophosphorylation independent of the upstream activating kinase MEK (*Figure 4*).

### Mutations in the β3-αC loop are cancer alleles in other kinase families

It has recently been discovered that changes in the β3-αC loop length of BRAF, EGFR, and HER2, none of which are in the CMGC group of kinases, also led to activation of these kinases (*Foster et al., 2016*). In these cases, the activating mutations shorten the loop rather than lengthen it. In these cases, the shortened loop may pull in the αC helix favoring the 'αC in' conformation. It is interesting that ERK has the opposite behavior. What is generally clear is that the length of the β3-αC loop is a crucial determinant of basal kinase activity.

In previous studies on the evolution of specificity within the CMGC kinase group, we found that an inferred ancestral DFG +1 mutation was also sampled in cancer mutations throughout the kinome (*Howard et al., 2014*), therefore it is possible that our evolutionary studies within the CMGC group of kinases may be more generally reflected throughout the kinome. The decreasing confidence in reconstructions with evolutionary distance makes it impossible to reconstruct an ur-kinase but features of this progenitor are certain to be retained across every kinase group. Thus, we speculate that β3-αC loop length was has been a critical modulator of basal kinase activity throughout kinase evolution.

Why, then, is ERK not a major oncogene? Oncogenic activation mutations in upstream kinases BRAF, MEK, EGFR and HER2 are frequently observed in cancer. Indeed, the Ras/RAF/MEK/ERK pathway is one of the most important signaling axes in oncogenesis. One possibility is that the upstream regulators have multiple targets in addition to ERK, and indeed this is certainly the case for Ras. Another intriguing possibility is that the dynamical properties of ERK signaling are crucial for its role in oncogenesis. ERK is not simply an ON/OFF switch, but rather oscillates or adapts in various contexts (*Albeck et al., 2013*). It will be interesting to leverage our mutants and investigate more recent inferred ancestral states that may have distinct dynamical properties to investigate this latter possibility.

### Evolutionary mutations synergize

We found that the evolutionary changes that place ERKs 1 and 2 under stringent regulation synergize in their effects. One change, a polar substitution in the 'gatekeeper' residue of the active site has been previously described as an activating mutation (*Emrick et al., 2006*; *Azam et al., 2008*). Evolution at the gatekeeper after AncERK1-5 proceeded along two divergent paths. The permissive methionine residue was retained along the branch that led to the modern ERK5 family, which also has high rates of autophosphorylation. The acquisition of a large C-terminal domain is thought to be an alternative source of regulation for ERK5 (*Buschbeck and Ullrich, 2005*). In ERK1-2 the gatekeeper was shifted to glutamine, a more restrictive state for autoactivation.

The second evolutionary change that we implicated in the decrease in autophosphorylation at the transition to AncERK1-2, a change in loop length between the β3 strand and αC helix, has not been reported before. The inferred deeper CMGC ancestors and the modern ERK kinases that have higher basal activity, ERK5 and ERK7, all had one extra amino acid in this loop (*Figure 3C* and *Figure 3—figure supplement 4*). The reversal of the shortening of this loop in AncERK1-2 and in

modern ERK1 and ERK2 drastically increased kinase activity (*Figure 3A*, *Figure 4* and *Figure 5—figure supplement 1*).

Each of these evolutionary changes confers an order of magnitude change in basal activity, and when combined, these two mutations increase the basal activity of ERKs 1 and 2 by more than two orders of magnitude. This additive, relatively independent effect of the two mutations may have provided evolutionary pathways that could have both conferred a selective advantage to intermediates and allowed for evolution towards a tightly controlled system without major fitness barriers.

## Structural dynamics and control of kinase activity

Our molecular dynamics simulations indicated that reversal of the inferred evolutionary mutations (i.e. reversion to a more active inferred ancestral state) had two clear effects on the rapid timescale (ns - µs) dynamics of ERK2. First, there was increased flexibility in the glycine rich loop and loop 16 (*Figure 5A–C*). The reorganization of loop 16 is important in the domain closure of kinases, which is an intrinsic part of the conformational shift to an active state. This observation is consistent with earlier MD simulations that showed that gatekeeper mutations can increase domain closure (*Barr et al., 2011*). However, our simulations are too short to directly observe domain closure. Markov State Models also suggested increased dynamics of the activation segment. The most compelling hypothesis suggested by our MD simulations was that the active site is more likely to attain a productive conformation in mutants, as indicated by more frequent formation of the K52-E69 salt-bridge in mutants than wild type (*Figure 5D*).

## Robust evolution in the CMGC kinase group

Previous ancestral reconstruction studies have highlighted the importance of epistasis in the diversification of function. For example, changes in the specificity of hormone-receptor ligand binding domains required stabilizing mutations (*Thornton et al., 2003*) prior to the advent of mutations that alter specificity, and the acquisition of novel mutations after acquisition of a new function made it difficult to revert to the inferred ancestral state (*Bridgham et al., 2009*). In our previous study on the evolution of kinase specificity, we found no requirement for stabilizing mutations – robust activity was maintained as specificity changed. In the current study, we find that the reversion of a single or double evolutionary mutation is also well tolerated and that we can revert modern ERK1 and ERK2 to an inferred ancestral state of high basal activity - there is not a strong "evolutionary ratchet". Together these observations suggest that kinases are quite tolerant of mutations, which perhaps explains their frequent dysregulation in diseases like cancer. The diversification of kinase specificity and regulation may also be underpinned by this surprising tolerance of a dynamic domain to mutations that drive neofunctionalization. Perhaps this 'evolvability' explains the prevalence of kinases as regulatory enzymes in *Eukaryotes*.

# Materials and methods

Key resources table

| Reagent type | Designation | Source | Identifiers | Additional information |
|---|---|---|---|---|
| Cell line | NIH3T3 | ATCC | CRL1658 | |
| Antibody | anti-ERK | Santa Cruz Bio | SC154 | WB:1:200 |
| Antibody | anti-p-ERK | Santa Cruz Bio | SC7383 | WB:1:200 |
| Recombinant DNA | His AncCM | This paper | pLH1347 | for purification from *E. coli* |
| Recombinant DNA | His AncMAPK | This paper | pLH1348 | for purification from *E. coli* |
| Recombinant DNA | His AncJPE | This paper | pLH1349 | for purification from *E. coli* |
| Recombinant DNA | His AncERK | This paper | pLH1350 | for purification from *E. coli* |

*Continued on next page*

*Continued*

| Reagent type | Designation | Source | Identifiers | Additional information |
|---|---|---|---|---|
| Recombinant DNA | His AncERK1-5 | This paper | pLH1571 | for purification from *E. coli* |
| Recombinant DNA | His AncERK1-2 | This paper | pLH1572 | for purification from *E. coli* |
| Recombinant DNA | His ERK1 | This paper | pLH1573 | for purification from *E. coli* |
| Recombinant DNA | His MEK | This paper | pLH1574 | for purification from *E. coli* |
| Recombinant DNA | MEK (without His tag) | This paper | pLH1575 | for purification from *E. coli* |
| Recombinant DNA | His ERK1-R84T | This paper | pLH1576 | for purification from *E. coli* |
| Recombinant DNA | His Elk1-GFP-FRB | This paper | pLH1577 | for purification from *E. coli* |
| Recombinant DNA | His AncCDK.MAPK-AxF | This paper | pLH1578 | for purification from *E. coli* |
| Recombinant DNA | His AncJPE-AxF | This paper | pLH1579 | for purification from *E. coli* |
| Recombinant DNA | His AncERK-AxF | This paper | pLH1580 | for purification from *E. coli* |
| Recombinant DNA | His AncERK1-2-Q89M | This paper | pLH1581 | for purification from *E. coli* |
| Recombinant DNA | His AncERK1-2-ins44R | This paper | pLH1582 | for purification from *E. coli* |
| Recombinant DNA | His AncERK1-2-ins44R-Q89M | This paper | pLH1583 | for purification from *E. coli* |
| Recombinant DNA | His Alt1.AncERK1-5 | This paper | pLH1584 | for purification from *E. coli* |
| Recombinant DNA | His Alt2.AncERK1-5 | This paper | pLH1585 | for purification from *E. coli* |
| Recombinant DNA | His Alt1.AncERK1-2 | This paper | pLH1586 | for purification from *E. coli* |
| Recombinant DNA | His Chimera 1 | This paper | pLH1587 | for purification from *E. coli* |
| Recombinant DNA | His Chimera 2 | This paper | pLH1588 | for purification from *E. coli* |
| Recombinant DNA | His Chimera 3 | This paper | pLH1589 | for purification from *E. coli* |
| Recombinant DNA | His Chimera 4 | This paper | pLH1590 | for purification from *E. coli* |
| Recombinant DNA | His Chimera 5 | This paper | pLH1591 | for purification from *E. coli* |
| Recombinant DNA | His AncERK1-5-del46R | This paper | pLH1592 | for purification from *E. coli* |
| Recombinant DNA | His AncERK1-5-M90Q | This paper | pLH1593 | for purification from *E. coli* |
| Recombinant DNA | His AncERK1-5-del46R-M90Q | This paper | pLH1594 | for purification from *E. coli* |
| Recombinant DNA | His ERK1-Q122M | This paper | pLH1595 | for purification from *E. coli* |
| Recombinant DNA | His ERK1-ins74N | This paper | pLH1596 | for purification from *E. coli* |

*Continued*

| Reagent type | Designation | Source | Identifiers | Additional information |
|---|---|---|---|---|
| Recombinant DNA | His ERK1-ins74N-Q122M | This paper | pLH1597 | for purification from *E. coli* |
| Recombinant DNA | His ERK1-ins74N AxY | This paper | pLH1598 | for purification from *E. coli* |
| Recombinant DNA | His ERK1-ins74N TxF | This paper | pLH1599 | for purification from *E. coli* |
| Recombinant DNA | His ERK1-ins74N AxF | This paper | pLH1600 | for purification from *E. coli* |
| Recombinant DNA | GST-ERK1-ins74N | This paper | pLH1601 | for purification from *E. coli* |
| Recombinant DNA | GST-ERK1- ins74N KD | This paper | pLH1602 | for purification from *E. coli* |
| Recombinant DNA | His ERK1-ins74W | This paper | pLH1603 | for purification from *E. coli* |
| Recombinant DNA | His ERK1-ins74v | This paper | pLH1604 | for purification from *E. coli* |
| Recombinant DNA | His ERK1-ins74a | This paper | pLH1605 | for purification from *E. coli* |
| Recombinant DNA | His ERK1-ins74G | This paper | pLH1606 | for purification from *E. coli* |
| Recombinant DNA | His ERK1-ins74Q | This paper | pLH1607 | for purification from *E. coli* |
| Recombinant DNA | His ERK1-ins74S | This paper | pLH1608 | for purification from *E. coli* |
| Recombinant DNA | His ERK1-ins74D | This paper | pLH1609 | for purification from *E. coli* |
| Recombinant DNA | His ERK1-ins74N-delK72 | This paper | pLH1610 | for purification from *E. coli* |
| Recombinant DNA | His ERK1-ins74N-delI73 | This paper | pLH1611 | for purification from *E. coli* |
| Recombinant DNA | His ERK1-ins74N-delS74 | This paper | pLH1612 | for purification from *E. coli* |
| Recombinant DNA | His ERK1-ins74N-delP75 | This paper | pLH1613 | for purification from *E. coli* |
| Recombinant DNA | His ERK1-ins74N-delF76 | This paper | pLH1614 | for purification from *E. coli* |
| Recombinant DNA | His ERK1-ins74N-delE77 | This paper | pLH1615 | for purification from *E. coli* |
| Recombinant DNA | His ERK1-ins74N-delH78 | This paper | pLH1616 | for purification from *E. coli* |
| Recombinant DNA | His ERK1-ins74N-delQ79 | This paper | pLH1617 | for purification from *E. coli* |
| Recombinant DNA | His ERK1-ins72N | This paper | pLH1618 | for purification from *E. coli* |
| Recombinant DNA | His ERK1-ins73N | This paper | pLH1619 | for purification from *E. coli* |
| Recombinant DNA | His ERK1-ins74N | This paper | pLH1620 | for purification from *E. coli* |
| Recombinant DNA | His ERK1-ins75N | This paper | pLH1621 | for purification from *E. coli* |
| Recombinant DNA | His ERK1-ins76N | This paper | pLH1622 | for purification from *E. coli* |

*Continued on next page*

*Continued*

| Reagent type | Designation | Source | Identifiers | Additional information |
|---|---|---|---|---|
| Recombinant DNA | His ERK1-ins77N | This paper | pLH1623 | for purification from *E. coli* |
| Recombinant DNA | His ERK1-ins78N | This paper | pLH1624 | for purification from *E. coli* |
| Recombinant DNA | His ERK1-ins79N | This paper | pLH1625 | for purification from *E. coli* |
| Recombinant DNA | ERK KTR | *Regot et al., 2014* | Addgene #59150 | Monitoring ERK activity in mammalian cells |
| Recombinant DNA | ERK1 wt DsRED | This paper | pLH1626 | for expresson in mammalian cells |
| Recombinant DNA | ERK1-ins74N DsRED | This paper | pLH1627 | for expresson in mammalian cells |
| Recombinant DNA | ERK1 wt overexpression | This paper | pLH1628 | for expresson in mammalian cells |
| Recombinant DNA | ERK1-ins74N overexpression | This paper | pLH1629 | for expresson in mammalian cells |
| Recombinant protein | MBP | EMD Millipore | 13–104 | kinase substrate |
| Recombinant DNA | His ERK2 | This paper | pLH1630 | for purification from *E. coli* |
| Recombinant DNA | His ERK2-ins55N | This paper | pLH1631 | for purification from *E. coli* |
| Recombinant DNA | His ERK2-Q103M | This paper | pLH1632 | for purification from *E. coli* |
| Recombinant DNA | His ERK2-ins55N-Q103M | This paper | pLH1633 | for purification from *E. coli* |

## Protein Expression and Plasmid Construction

pET28b vectors were used for bacterial expression. Open Reading Frames (ORFs) of the Erk proteins were fused N terminally to the 6X histidine tag for purification. Glutathione S Transferase (GST) was inserted between the 6X His-tag and the Erk ORF in order to increase protein molecular weight. For site-directed and domain-swapping mutagenesis, primers containing the point mutation sites or domain junction sites for chimera protein were designed. The PCR fragments and the vectors were assembled according to the manufacturer's instructions using Gibson Assembly Master Mix kit from New England Biolabs (NEB). All the constructs were verified via DNA sequencing.

## Mammalian cell culture

NIH3T3 ATCC CRL1658 Cell lines were verified using short tandem repeat DNA profiling (STR Profiling) by PCR amplification of 9 STR loci plus Amelogeninusing Promega GenePrint 10 System, Fragment Analysis with ABI 3730XL DNA Analyzer, comprehensive data analysis with ABI Genemapper software and final verification using supplier databases including ATCC and DSMZ. Cell lines were checked and tested negative for mycoplasma infection. Cells were cultured in Dulbecco's Modified Eagle's Medium (DMEM) supplemented with 10% Fetal Bovine Serum (FBS), penicillin, and streptomycin. The cells were incubated at 37°C and 5% $CO_2$. Transfections of HEK293T cells were performed using TransIT-293 Transfection Reagent (Mirus), according to the manufacturer's instructions. Viral supernatants were collected 48–72 hr following transfection of 293 T cells. After virus infection, NIH3T3 cells were selected with 1–2 μg/ml puromycin. For western blot, cells were first washed three times with PBS, then incubated in medium with no serum and treated with 2 μM Trametinib. After 12 hr, cells were lysed by addition of SDS loading buffer containing 20% glycerol, 4% SDS, 0.1 M Tris (pH 6.8), 0.2 M dithiothreitol (DTT), and phenol blue dye, followed by boiling at 100°C for 10 min.

## Protein purification

Protein purification from *E. coli* cells was performed as previously described (*Howard et al., 2014*). Briefly, proteins were expressed in Rosetta2 DE3 competent cells by induction with 100 µM IPTG for 18 hr at 16˚C. Bacterial culture were collected and centrifugated at 4000 rpm for 20 min at 4˚C. The cell pellet was resuspended in cold lysis buffer (50 mM NaH2PO4, 300 mM NaCl, 10 mM imidazole pH7.6, 1 mM PMSF). After sonication, the lysate was centrifuged at 12000 rpm for 20 min at 4˚C. The supernatant was mixed with magnetic Ni-NTA beads (Biotool) and incubated for 1 hr at 4˚C. The bound beads were collected and rinsed 3 × with wash buffer containing (50 mM NaH2PO4, 300 mM NaCl, 40 mM imidazole pH7.6). The bound proteins were eluted with elution buffer (50 mM NaH2PO4, 300 mM NaCl, 500 mM imidazole pH7.6). The elution was dialyzed into storage buffer (150 mM NaCl, 50 mM HEPES pH8, 1 mM DTT, 10% glycerol) using Zeba Spin Desalting Columns (Thermo scientific), and flash frozen in aliquots with liquid nitrogen and stored at 80˚C.

## Western blots

For recombinant proteins, 200 ng protein were mixed with SDS loading buffer and boiled at 100˚C for 10 min. For mammalian cells, the protein lysates were prepared by removing the medium, washing with PBS, and adding SDS buffer directly to the plate. Cell lysates were collected and boiled for 10 min. The samples were separated on 4–12% Bis-Tris gels (ThermoFisher Scientific) and transferred to a nitrocellulose membrane. Membranes were probed with anti-ERK (SC154, Santa Cruz Biotechnology) or anti-p-ERK (SC7383, Santa Cruz Biotechnology).

## In vitro kinase assays

0.05 µg purified recombinant 6x His-tagged Erk protein was used in a total volume of 10 µl (approximately 0.1 µM). Final reaction conditions were 20 mM Hepes, pH 7.5, 10 mM MgCl$_2$, 50 mM NaCl, 0.75 µg/µl myelin basic protein (MBP), 100 µM ATP, and 0.05 µCi of [γ-$^{32}$P]ATP. The kinase reactions proceeded for 30 min at room temperature and were terminated by addition of 6 µl 5x SDS loading buffer (10% SDS, 0.5 M DTT, 50% glycerol, 0.25% Bromophenol blue). All samples were separated on 4–12% Bis-Tris gels (ThermoFisher Scientific). The gel was dried and exposed to a phosphor screen. Phosphor screens were analyzed with a Typhoon 9500 scanner (GE) using ImageQuant software (GE). Michaelis–Menten curves were generated in a similar manner, except the buffer contained 300 µM ATP and varying concentration of MBP. Data were fit by nonlinear regression to the Michaelis–Menten model $V_0 = [ERK]* k_{cat} *[S]/K_M+[S]$ using Prism 8 (GraphPad software) and the fit was used to determine values for $k_{cat}$ and $K_M$.

## Autophosphorylation assay

0.1 µg purified recombinant 6x His-tagged Erk protein (approximately 0.2 µM) was incubated in kinase reaction buffer (20 mM Hepes, pH 7.5, 10 mM MgCl$_2$, 50 mM NaCl, 25 µM ATP, and 0.05 µCi of [γ-$^{32}$P]ATP) without substrate. The autophosphorylation reactions proceeded for 30 min at room temperature and were terminated by addition of 6 µl 5x SDS loading buffer. All samples were separated on 4–12% Bis-Tris gel (ThermoFisher Scientific). The gel was dried and exposed to phosphor screen. Phosphor screens were analyzed with a Typhoon 9500 scanner. Autophosphorylation of ERK1 ins74N was measured at 5, 10, 15, and 20 min, as previously desctibed (*Emrick et al., 2006*). Reactions were terminated, and proteins were seperated by SDS/PAGE, and phosphate incorporation was quantified by using ImageQuant software (GE).

## Liquid chromatography MS

Sample preparation for mass spectrometry analysis

50 µg of each purified protein was reduced using dithiothreitol (2 µl of 0.2 M) for 1 hr at 55˚C. The reduced cysteines were subsequently alkylated with iodoacetamide (2 µl of 0.5 M) for 45 min in the dark at room temperature. The protein samples were digested with trypsin (Promega) at a 100:1 (protein:enzyme) ratio overnight at room temperature. The pH of the digested protein lysates were lowered to pH < 3 using trifluoroacetic acid (TFA). The digested lysates were desalted using C18 solid-phase extraction (Sep-Pak, Waters). 40% acetonitrile (ACN) in 0.5% acetic acid followed by 80% acetonitrile (ACN) in 0.5% acetic acid was used to elute the desalted peptides. The peptide eluate was concentrated in the SpeedVac and stored at −80 C.

## Mass spectrometry analysis

1 µg of each sample was loaded onto a trap column (Acclaim PepMap 100 pre-column, 75 µm × 2 cm, C18, 3 µm, 100 Å, Thermo Scientific) connected to an analytical column (EASY-Spray column, 50 m × 75 µm ID, PepMap RSLC C18, 2 µm, 100 Å, Thermo Scientific) using the autosampler of an Easy nLC 1000 (Thermo Scientific) with solvent A consisting of 2% acetonitrile in 0.5% acetic acid and solvent B consisting of 80% acetonitrile in 0.5% acetic acid. The peptides were eluted into either an Orbitrap QExactive (Thermo Fisher Scientific) or Orbitrap QExactive HF-X mass spectrometer (Thermo Fisher Scientific) respectively, using the following gradient: 5–35% in 60 min, 36–45% in 10 min, followed by 45–100% in 10 min. Protein samples for Erk1 and Erk1-ins74N were acquired on an Orbitrap QExactive mass spectrometer, with full MS spectra recorded at a resolution of 70,000, an AGC target of 1e6, with a maximum ion time of 120 ms, and a scan range from 400 to 1500 m/z. Following each full MS scan, twenty data-dependent MS/MS spectra were acquired. The MS/MS spectra were collected with a resolution of 17,500 an AGC target of 5e4, maximum ion time of 120 ms, one microscan, 2 m/z isolation window, fixed first mass of 150 m/z, dynamic exclusion of 30 s, and normalized collision energy (NCE) of 27. Protein samples for AncCDK.MAPK, AncMAPK, AncJPE, AncERK, AncERK1-5 and AncERK1-2 were acquired on the QExactive HF-X using data dependent acquisition mode (DDA) or parallel reaction monitoring mode (PRM). In both cases, high resolution full MS spectra were recorded with a resolution of 45,000, an AGC target of 3e6, a maximum ion time of 45 ms, and a scan range from 400 to 1500/z. For samples acquired using DDA mode, MS/MS spectra were collected with a resolution of 15,000 an AGC target of 1E5, maximum ion time of 120 ms, one microscan, 2 m/z isolation window, fixed first mass of 150 m/z, dynamic exclusion of 30 s, and normalized collision energy (NCE) of 27. For samples acquired using PRM mode, PRM scans were acquired for the peptides of interest following each full MS scan. The MS/MS spectra were collected with a resolution of 15,000 an AGC target of 2E5, maximum ion time of 200 ms, one microscan, 2 m/z isolation window, fixed first mass of 150 m/z, dynamic exclusion of 30 s, and normalized collision energy (NCE) of 27.

## Data processing

For all samples, the MS/MS spectra were searched against a given protein sequence of interest using Byonic (Protein Metrics Inc, San Carlos, CA)(*Bern et al., 2012*). The mass tolerance was set to 10ppm for MS1 and 0.02 Da for MS2 searches. False discovery rate (FDR) of 1% was applied on the protein level. The Byonic search included fixed modification of carbamidomethylation on cysteine and variable modifications of oxidation on methionine, and phosphorylation at serine, threonine and tyrosine residues. The spectra of the peptides of interest were manually verified to localize the phosphorylation site. For relative quantification of the phosphorylation sites the semi-automatic program Byologic (Protein Metrics Inc, San Carlos, CA) was used to obtain the area under the curve (AUC) for the extracted ion chromatogram of each peptide isoform. The AUC was manually verified and adjusted if appropriate for each isoform. The percent phosphorylation was obtained by dividing the isoform AUC by the total AUC for the peptide. The data can be found in the supplementary data table MS-source.

## Imaging and image analysis

To assess the activity of Erk in mammalian cells, ERK-KTR was expressed from the CMV promoter in vector pLentiCMV (Addgene #59150). ERK kinase was expressed along with a dsRed reporter expressed from an IRES. Cells were seeded on six well plates one day before imaging. At 30% confluency, media was changed to media without FBS and containing Trametinib (final concentration 2 µM) 2 hr prior to imaging. Cells were imaged with a Nikon Eclipse Ti fluorescence microscope. All imaging was carried out at constant conditions of 37°C, 5% $CO_2$, and constant humidity. The subcellular distribution of the KTR reporter was analyzed with Image J. Cells expressing both Kinase and KTR reporter were used for analysis.

## MD simulations

To perform MD simulations of ERK2-ins55N, ERK2-Q103M (gatekeeper mutation), and ERK2-ins55N + Q103M (double mutant) mutants, we used the unphosphorylated ERK2 (PDB ID: 1ERK). We introduced the insertion to ERK2 (55N) using Discovery Studio 4.5 software (*Biovia, 2015*), and

then modeled the loops using Modloop web server (*Fiser et al., 2000*). We mutated the gatekeeper residue 103(Q > M) in ERK2 and ERK2-ins55N using the VMD Mutator tool to obtain the structures.

We next solvated the proteins with water in cubic boxes with at least 1 nm padding on all sides of the protein and neutralized with a minimal amount of NaCl using OpenMM 7.2 (*Eastman et al., 2017*). We recorded the largest number of water and ion molecules added to any of the proteins (20,959 molecules) and solvated again in cubic boxes using that number of molecules. This resulted in systems of between 68,670 and 68,684 atoms. We then energy minimized and equilibrated the systems for 10 ns in the NpT (p=1 atm, T = 300 K) ensemble using the OpenMM Langevin integrator with a two fs timestep and a 20 ps$^{-1}$ collision rate. We controlled the pressure by a Monte Carlo molecular-scaling barostat with an update interval of 50 steps. We treated non-bonded interactions with the Particle Mesh Ewald (*Darden et al., 1993*) method using a real-space cutoff of 0.9 nm and relative error tolerance of 0.0005, with grid spacing selected automatically. We constrained all covalent bonds involving hydrogen and did not remove the center of mass motion since this was unnecessary given the use of a Langevin integrator.

Next, we packaged these simulations as seeds with OpenMM 6.3.1 (*Eastman et al., 2013*) to initiate massively parallel distributed MD simulations (production simulations) on Folding@home (*Shirts and Pande, 2000*). For all simulations, we used the parameter files included in the OpenMM 7.2 distribution (*Eastman et al., 2017*) for the Amber ff99SB-ILDN force field (*Lindorff-Larsen et al., 2010*) and the TIP3P rigid water model (*Jorgensen et al., 1983*). For NaCl, we used the adapted Aqvist (Na$^+$) (*Aqvist, 1990*) and Smith and Dang (Cl$^-$) (*Smith and Dang, 1994*) parameters. For production simulations, we used the same Langevin integrator as the NpT equilibration described above, except that to provide realistic heat exchange with a thermal bath while minimally perturbing dynamics, we set the Langevin collision rate to one ps$^{-1}$. In total, we generated 1000 independent MD simulations on Folding@home (*Shirts and Pande, 2000*) for each system, which totaled 4,000 MD simulations, of which 3999 successfully collected data. We propagated the simulations until a total of at least 500 µs of simulation time for each system had been collected, resulting in the grand total of 2.05 ms of simulation time and 4,104,841 frames (see *Figure 5—figure supplement 3* for trajectory length distribution histogram). The average trajectory length was 513 ns. This amount of simulation time corresponded to ~240 GPU-years on an NVIDIA GeForce GTX 980 processor. To use conformational snapshots (frames) in subsequent analyses, we stored them at intervals of 0.5 ns/frame.

## Calculation of RMSFs

To study the flexibility of the kinase, we calculated root mean square fluctuations (RMSFs) for each trajectory using NumPy (*Oliphant, 2006*). These calculations measure the amplitude of residue C$\alpha$ atom motions during MD simulations. To allow for further relaxation from the initial equilibrated structures, we did not use the first 50 ns of each trajectory. We used Seaborn (mwaskom/seaborn) with Matplotlib (*Hunter, 2007*) to plot the RMSF means over all trajectories, 95% confidence intervals of the means, and differences between the wild type and mutant (*Figure 5A–C*).

## Calculation of domain closure

To study the motion between N and C domains we calculated the domain closure as the angle between the helices C and E. We did a least squares fit of a vector ('helix axis') for each helix to the positions of the carbonyl C and O atoms, and then recorded the angle between those vectors (*Figure 5G*). To allow for further relaxation from the initial equilibrated structures, we did not use the first 50 ns of each trajectory.

## Calculation of residue-residue distances

To understand the mutation-induced changes in the interactions between residues that are important for activity and/or along the regulatory and catalytic spines, we calculated the closest heavy atom residue-residue distances using all frames using MDTraj (*McGibbon et al., 2015*) for each trajectory. Again, we did not use the first 50 ns of each trajectory. Specifically, in ERK2, we calculated the residue-residue distances between: 1) Catalytic sites (E69-F166, K52-E69, D165-K52) 2) Regulatory spine (I84-L73, L73-F166, F166-H145, H145-D208) and 3) Catalytic spine (V37-A50, A50-L154, L153-L155, L153-M106, L155-M106, M106-I215, I215-M219). To determine whether the probability

distributions of the distances were significantly altered due to mutation, we performed the Kolmogorov-Smirnov (K-S) test with SciPy (*Jones et al., 2001*) using the K-S statistic cutoff of 0.1. We plot the probability distributions of distances where at least one of the mutants showed difference from that of the WT using Seaborn with Matplotlib (*Hunter, 2007*) (*Figure 5A–F*). Analogically, to understand the mutation-induced changes in the interactions between residues proposed (*Emrick et al., 2006*) to be important in controlling accessibility of conformations able to autophosphorylate the activation lip, we calculated the following closest heavy atom residue-residue distances: V186-L168, T188-D147, T188-K149, Y185-D147.

## Calculation of contact maps
We calculated the closest heavy atom residue-residue distances for all pairs of residues for each trajectory using MDTraj (*McGibbon et al., 2015*), converted to binary contact maps using a contact cutoff of 4 Å, and averaged over all frames of the trajectory using NumPy (*Oliphant, 2006*). The first 50 ns of each trajectory were not used to allow for further relaxation from the initial equilibrated structures. The contact maps for all trajectories of a given system were averaged. The wild type contact maps were subtracted from the mutant contact maps to generate differential contact maps for further annotation (*Figure 5—figure supplement 2*).

## Construction of Markov state models (MSMs)
We trained the XGBoost (*Chen and Guestrin, 2016*) gradient boosted decision tree algorithm on the per-trajectory mean binary contacts to classify the origin of a trajectory from the four protein systems. Following an established method (*Brandt et al., 2018*) using an iterative exclusion principle, we trained and cross-validated (2/3 of the dataset were used for training, and 1/3 for testing) the model, then removed the contact feature with the highest importance to classification and repeated the procedure with remaining features. This was continued until the classification accuracy dropped below 90%, resulting in 125 top contact features. The distances corresponding to those contacts were calculated and used as features for Markov state models (MSMs). All modeling was performed using PyEMMA (*Scherer et al., 2015*). First, we commute-mapped (*Noé et al., 2016*) and reduced the dimensionality of the data using time-lagged independent component analysis (tICA; *Pérez-Hernández et al., 2013*), keeping 95% of the kinetic variance, resulting in X tICs. The data were then clustered into 100 microstates using k-means, to partition the data into a minimal number of states sufficient for reasonable description of the dynamics, while avoiding overfitting (*Husic et al., 2016*). There was negligible overlap in state space between WT/Q103M and the insertion containing 55N/55N-Q103M – the former occupied 30 microstates and the latter occupied 71 microstates, with only one microstate in common, likely to be due to noise. We hence built two separate MSMs – one for WT and Q103M combined, the other for 55N and 55N-Q103M combined. The MSMs were built from two different systems so that we could find a common definition of macrostates, under the assumption that the Q103M mutation causes changes to the populations of macrostates but not their metastability. We assessed the implied timescales of the MSMs, and found approximate convergence at 100 ns lag time. The top timescales were 0.47 μs and 1.81 μs for the WT/Q103M and 55N/55N-Q103M MSMs, respectively. We sought to coarse-grain the MSMs into less than five macrostates for easy interpretability, and we chose their number so that the sum of population differences between WT and Q103M, or 55N and 55N-Q103M was maximized for best interpretability of differences caused by the mutation. This resulted in 3 and 4 macrostates for the WT/Q103M and 55N/55N-Q103M MSMs, respectively. The coarse graining was performed using PCCA++. The MSMs were validated with Chapman-Kolmogorov (C-K) tests using those macrostate definitions. To calculate the populations of the macrostates in each individual system, four separate MSMs were also built. The choice of the lag time and model fidelity were again evaluated with implied timescales and C-K tests. Finally, the macrostate populations were calculated by weighted sum over microstate populations in each of the four individual MSMs, using the PCCA ++fuzzy memberships of microstates to macrostates from the 2 MSMs built with combined systems as weights. We randomly drew 100 samples from the dataset for each macrostate and calculated their mean contact maps. As a reference, we calculated the average contact map for WT weighted by macrostate populations, and subtracted it from the mutant macrostate contact maps to generate differential contact maps for further annotation (*Figure 5—figure supplements 4* and *5*).

## Data and code accessibility

The molecular dynamics simulation data is available via the Open Science Framework at: https://osf.io/dp4cb/.

The code used to analyze the data is available at https://github.com/choderalab/ERK-ancestral-materials (*Wiewiora, 2019*; copy archived at https://github.com/elifesciences-publications/ERK-ancestral-materials).

## Ancestral reconstruction of the ERK1 and ERK2 lineage

We identified orthologs of human ERK1, ERK2, ERK5 and ERK7 using a protein BLAST (*Altschul et al., 1990*) search against the genomes of a diverse set of eukaryotes. The obtained sequences were clustered using CD-HIT (*Li and Godzik, 2006*) at 95% identity to collapse closely related kinases into one sequence. These were then reverse blasted against the human ERKs to identify the kinase that they were most closely related to. For the purposes of reconstruction, they were considered to be the ortholog of this kinase. After some manual curation, the final list of 150 sequences was fed into Phylobot (*Hanson-Smith and Johnson, 2016*) for ancestral reconstruction with sequences from similarly collected p38 and JNK orthologs used as an outgroup to root the tree.

All primary data can be accessed at the following urls:

InputFasta File:

http://webdoc.nyumc.org/phylobot/Sang2019_Input_fasta_ERK_resurrection.fasta

Output Fasta File:

http://webdoc.nyumc.org/phylobot/Sang2019_Output_fasta_ERK_ancestors.fasta

SQL Database containing all resurrection data:

http://webdoc.nyumc.org/phylobot/Sang2019_ERK_ASR.db

The full reconstruction can be found at:

http://phylobot.com/350478581/msaprobs.PROTCATLG/ancestors

## Acknowledgements

We thank Conor Howard for initial experiments and contributions to this project, David Truong for help writing the manuscript; Greg Brittingham and Morgan Delarue for help with imaging and analysis; Laura Hug and Victor Hanson-Smith for advice regarding ancestral reconstruction; David Engelberg for sharing plasmids; and Sezen Vatansever for setting up the initial protein structures. We also thank Yu Zhao and the rest of the Holt lab for helpful discussions regarding the project. We also thank our core facilities, particularly Beatrix Uberheide and Avantika Dhabaria at the proteomics core at NYU Medical Center. The Mass Spectrometric experiments were in part supported by the NYU School of Medicine and the Laura and Isaac Perlmutter Cancer Center Support grant P30CA016087 from the National Cancer Institute. We gratefully acknowledge funding from the William Bowes Fellows program and the Vilcek Foundation (LJH). RPW gratefully acknowledges funding from DoD Horizon Award W81XWH-17-1-0412; JDC from NIH grants R01GM121505 and P30CA008748; and ZHG from LUNGevity foundation and start-up funds from Icahn Institute for Data Science and Genomic Technology.

## Additional information

### Funding

| Funder | Grant reference number | Author |
| --- | --- | --- |
| Vilcek Foundation | | Dajun Sang<br>Sudarshan Pinglay<br>Liam J Holt |
| National Cancer Institute | P30CA016087 | Dajun Sang<br>Sudarshan Pinglay<br>Liam J Holt |
| U.S. Department of Defense | W81XWH-17-1-0412 | Rafal P Wiewiora |

| National Institute of General Medical Sciences | R01GM121505 | John D Chodera |
|---|---|---|
| National Cancer Institute | P30CA008748 | John D Chodera |
| National Institute of General Medical Sciences | GM104047 | Hua Jane Lou Benjamin E Turk |

The funders had no role in study design, data collection and interpretation, or the decision to submit the work for publication.

### Author contributions
Dajun Sang, Conceptualization, Formal analysis, Investigation, Methodology, Writing—original draft; Sudarshan Pinglay, Conceptualization, Data curation, Software, Formal analysis, Validation, Investigation, Visualization, Writing—review and editing; Rafal P Wiewiora, Data curation, Formal analysis, Investigation, Visualization; Myvizhi E Selvan, Data curation, Software, Formal analysis, Investigation, Visualization, Methodology, Writing—review and editing; Hua Jane Lou, Formal analysis, Investigation, Writing—review and editing; John D Chodera, Zeynep H Gümüş, Conceptualization, Data curation, Software, Formal analysis, Supervision, Visualization, Methodology, Writing—review and editing; Benjamin E Turk, Conceptualization, Data curation, Formal analysis, Funding acquisition, Investigation, Writing—original draft, Writing—review and editing; Liam J Holt, Conceptualization, Data curation, Formal analysis, Funding acquisition, Investigation, Visualization, Writing—original draft, Writing—review and editing

### Author ORCIDs
Dajun Sang https://orcid.org/0000-0002-7512-074X
Sudarshan Pinglay https://orcid.org/0000-0002-8781-1476
Rafal P Wiewiora http://orcid.org/0000-0002-8961-7183
John D Chodera http://orcid.org/0000-0003-0542-119X
Zeynep H Gümüş http://orcid.org/0000-0002-7364-2202
Liam J Holt https://orcid.org/0000-0002-4002-0861

### Decision letter and Author response
Decision letter https://doi.org/10.7554/eLife.38805.sa1
Author response https://doi.org/10.7554/eLife.38805.sa2

## Additional files

### Supplementary files
• Source data 1. Raw data for LC-MS quantification of phosphrylation level of 'TEY' motif in inferred ancestral and modern kinases.

• Transparent reporting form

### Data availability
All data generated or analysed during this study are included in the manuscript and supporting files.

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
