## [Decision Letter]

Thank you for submitting your article "Ancestral resurrection reveals mechanisms of kinase regulatory evolution" for consideration by *eLife*. Your article has been favorably reviewed by three peer reviewers, and the evaluation has been overseen by John Kuriyan as the Senior Editor and the Reviewing Editor. The following individual involved in review of your submission has agreed to reveal her identity: Dorothee Kern (Reviewer #3).

The reviewers have discussed the reviews with one another and the Reviewing Editor has drafted this decision to help you prepare a revised submission.

In this paper, Sang et al. use ancestral reconstruction to study the evolution of MAP kinase regulation. The authors analyzed the specificity and activity of several kinases that represent common ancestors of major kinase families within the CMGC group. Interestingly, they found that the dependence on upstream MAPKK phosphorylation occurred at the transition between the common ancestor of the ERK 1/2/5 kinases (AncERK1-5) and the common ancestor of ERK 1/2 (AncERK1-2). Specifically, the authors present convincing data that a two amino acid change late in evolution between the ancestor of ERK5 and ERK1 and 2 toward ERK1-2 caused a large decrease in autophosphorylation rate, and autophosphorylation was replaced by MEK phosphorylation. This is a beautiful result!

Overall, the study is well designed and presented, and the results will be of high interest to investigators studying kinase regulation.

The manuscript builds upon the authors' previous *eLife* paper, which described the divergence of substrate specificity in related kinases using similar approaches, and thus warrants publication as an *eLife* Research Advance after revision.

Major issues that must be addressed, with additional experiments as necessary:

Although this is broken down into separate points, 1 to 6 below, all of these issues are related to putting the activities of the relevant kinases on a more rigorous footing. We ask that a substantial response be prepared for these questions, but the authors should judge whether particular experiments can be done in a realistic timeframe or not. Note that these questions are important, so a simple response that none of the experiments can be done in a timely fashion will not be deemed a satisfactory response.

1) The authors use MBP phosphorylation as readout for intrinsic activity, but as in their previous paper, they show that each ancestral kinase has a slightly different substrate specificity. Is it possible that the differences observed for MBP phosphorylation are due to difference in sequence specificity, rather than intrinsic activity?

2) In this assay, does MEK have any significant activity toward MBP? This should be measured as a control.

3) The activity measurements done in the paper are endpoint assays of myelin basic protein phosphorylation. For at least the key proteins (ERK1, AncERK1-5, AncERK1-2), the authors should carry out kinetic experiments. This would allow them to determine whether the effects are driven by changes in k_cat_, K_M_, or a combination.

4) What are the phosphorylation states of each protein prior to the MBP phosphorylation reactions? It would be important to compare the activities of all proteins for the same phosphorylation states (nonphosphorylated versus doubly phosphorylated proteins). The activity data presented in Figure 1B represent the results of different mixtures of non- and phosphorylated kinases according to Figure 2B and C.

For this, the quantification of phosphorylation for all samples need to be measured first, and then activity data collected for clean species (in Figure 2B and C, only a subset of proteins were characterized). Particularly AncERC1-5and AncRC1-2 are missing in respect to analysis of differential phosphorylation rates, the crucial proteins where the switch seem to happen in evolution.

5) A distinction should be made throughout the manuscript between "intrinsic activity" and regulation by activation loop auto-phosphorylation. Differences in "intrinsic activities" would ideally be assessed by comparing the enzymatic activities of all kinases in the same phosphorylation states. In other words, how active are each of the unphosphorylated kinases, and how active are those same kinases after activation loop phosphorylation? What the authors are measuring in the beginning of the Results section, and in Figure 1B, appears to be the consequence of a difference in the ability to auto-phosphorylate and thus auto-activate, not necessarily a difference in intrinsic activity.

6) Related to these points, it would be informative to see raw data (phosphorimages) from the MBP phosphorylation reactions in Figure 1B as a figure supplement, rather than just a normalized quantification of endpoint intensities.

7) Finally, we ask that the amino acid sequences of all proteins (modern and resurrected) be included in the supplementary materials, and it would be of value to give the number of substitutions in the text. The full tree used for ancestral sequence reconstruction (ASR), including the species, needs to be included for reproducibility for the field. A supplementary figure that shows the probabilities of the nodes, and information of the posterior probabilities for the full sequences is essential to evaluate ASR.

Other important issues that require response:

1) The authors should avoid language that implies that the ancestral protein sequence is actually known. A sophisticated reader will understand what is meant, but we are concerned that the way that much of the paper is written a strong impression is given that the ancestral sequence is actually synthesized. This is not just a semantic point, because it may turn out that errors in the sequence reconstruction have led to misleading results. We do not consider this point to invalidate the conclusions of this study, but we ask that the language used to describe the "ancestral" sequences be changed throughout the manuscript. A few comments, given below, should help explain what we mean.

The method of ancestral sequence reconstruction is a powerful approach to dissect molecular differences between close-related biomolecules. However, the use of the dramatic term "resurrection" should be avoided throughout this manuscript, as it implies that we have explicit knowledge of the true protein sequences found in ancestral organisms, which is then "resurrected". The more common term, "reconstruction", is more appropriate, as it indicates that a sequence model of a plausible ancestral protein is being generated using the best available evidence. An example of a statement that implies that the ancestor is truly known, and that the authors should avoid is the following:

"As part of this study, we observed that a deep ancestor of this group had relatively high intrinsic activity even in the absence of any activating kinase of or allosteric activators."

In the beginning of the main section of the paper, the authors use the following phraseology, which correctly captures the inherent uncertainty in the "resurrected" sequences: "We synthesized the coding sequences for these inferred ancestors".

2) The manuscript is focused on the specific late evolution between ERK5 and ERK 1 and 2, and there is no information about the evolution of regulation towards CDKs, MAPKs, *JNK*, p38. Therefore the Introduction, Abstract and title should be modified to reflect the data from this manuscript (for instance: "Ancestral resurrection reveals mechanism of kinase regulatory evolution of ERK").

3) To enlighten the reader about the evolutionary trajectory, key information to add would be how modern ERK5, ERK 7 that branch off the ancestors used in this study are behaving in respect to activities of their nonphosphorylated and phosphorylated activation loop states towards substrates, and their ability to autophosphorylate.

4) What is the explanation of a much higher activity of AncERK1-2 with the two mutations in the presence of MEK relative to AncERK1-2 in the presence of MEK? Could the authors also include the data for the reverse experiment, AncERK1-5 M89Q 44Rdel, both in the absence and presence of MEK, in Figure 3?

5) 300 ns trajectories are very short MD simulations, and it is quite unlikely that refolding of the L16 loop coupled to domain closure is on that time scale. NMR data for ERK has suggested that these processes occur on the millisecond time-scale. Thus, the MD simulations are not convincing due to the discrepancy in time scale between the suggested conformational changes and the length of the simulations, and rather weaken the manuscript than strengthen it. The authors should consider carefully what information is contributed by the MD simulations, and then describe that with appropriate cautionary notes. Alternatively, remove the MD simulations from the manuscript.

Should the MD simulations be retained, the authors should clarify what atoms are being used to measure the distances between residues in the analyses of MD simulations. In particular, for salt bridges, as in the one analyzed in Figures 5C, F, it would be most informative to look at distances between terminal or near-terminal atoms of the side chains (e.g. the terminal nitrogen of the lysine sidechain and the δ carbon or one of the oxygens of the glutamate residue).

6) The figures could be better arranged and assembled, it is hard to jump back and forth between gels and activity figures. Could all data be presented in a quantitative format (activity towards MBP and autophosphorylation)?

7) The authors should include an LC/MS quantification of TEY motif phosphorylation for AncERK1-5 and AncERK1-2 (i.e., a similar analysis as was done in Figure 2C).

8) Figure 4E does not conclusively establish the mechanism of autophosphorylation; His6-ERK1 ins74N could still be undergoing intermolecular (trans) autophosphorylation. This point would need to be established by kinetic experiments, e.g. by measuring the concentration dependence of autophosphorylation.

9) Figure 6D: the authors should present a quantification of the Western blots (the ratio of phosphorylated ERK1 to total ERK1).

10) The activation loops of the ancestral kinases differ in length and sequence. Notably, the ancestral CDK.MAPK kinase has a -3 Arg residue, for which this kinase still has some residual preference. If canonical sequence-specificity is not relevant because of a cis-autophosphorylation mechanism, it is still worth noting that the sequence of the activation loop could affect activation loop dynamics. The authors should discuss how the composition of activation loop itself might impact their findings. This is somewhat addressed by the chimera experiments but would still be worth mentioning.

[Editors' note: further revisions were requested prior to acceptance, as described below.]

Thank you for resubmitting your work entitled "Ancestral reconstruction reveals mechanisms of ERK regulatory evolution" for further consideration at *eLife*. Your revised article has been favorably evaluated by John Kuriyan (Senior Editor and Reviewing Editor), and three reviewers. The manuscript is now, in principle, suitable for publication in *eLife*, but the reviewers have identified some remaining issues that need to be addressed before acceptance, as outlined below. Please use your best judgement in revising the manuscript with reference to these points, and we will make a final decision as quickly as possible. When submitting the revised manuscript, please provide a point-by-point response to the review comments. Also, please add figure numbers to each individual figure in the submitted manuscript, and include page numbering on each page of the manuscript – the lack of such numbering introduced an element of confusion in the review of the previous manuscript.

The premise of this study, that ancestral sequence reconstruction can be used to identify molecular mechanisms by which kinase regulation evolved, is quite intriguing. The authors do a nice job of exploring this premise within the MAP kinase family. The results are novel and of broad interest to the kinase field. Furthermore, the authors have adequately addressed most of the concerns raised after the original submission, although a few remaining points require attention before acceptance.

1) A major issue is that for AncERK, AncJPE, AncMAPK, AncCDK.MAPK (which are evolutionary further from the modern proteins), still no posterior probabilities are provided. They need to be shown, as previously asked by the review. In addition, no experiments were performed to test for their phosphorylation status, and no experiments were done on their modern counterparts. Only activity was measured for these ancestors (Figure 1A), but their relation in respect to modern CDK, MAPK, *JNK*, p38 is not discussed either in the manuscript. Without the proper analysis on those ancestors the authors may wish to remove them and to keep the focus on the ERK ancestors. The latter is the story of this manuscript, and this part is interesting and convincing.

2) The manuscript still does not correctly describe the quantification of the percentages of non-phosphorylated, single and double phosphorylated (TxY motif) proteins used in the activity studies. Figure 2—figure supplement 1B is provided, but one cannot follow what is exactly shown, and how these data can be used for quantification of phosphorylation. In the text it is described as "more" or "less" phosphorylation. Since differences in autophosphorylation is the center of the findings of this manuscript, this needs to be quantified accurately. Are these full-length proteins, or peptides shown in this figure, and the results of what kind of experiment are shown? Even ERK 1 appear to have some phosphorylation. What does "modified TxY " mean?

3) The part of the manuscript pertaining to Point No. 4 in the "Other Important Points" in the original review, still needs further clarification in the manuscript. If this result is puzzling and currently cannot be explained unambiguously with a model, it is important to articulate this for the reader.

4) The MD simulations are done on ERK2 (Figure 5), but all the experiments in the manuscript are performed on ERK1 (Figure 4). Why is this?

There is difficulty in seeing how the MD figures provided now help explain why the mutants of ERK1 autophosphorylate the TxY motive in cis much more efficiently than wtERK1. In Figure 5 all proteins have a bimodal distribution for the plotted distances (which are slightly altered). If one of these two substates is interpreted to be the active conformation, one would not expect the experimental result of a big difference in phosphorylation state between these proteins. The authors correctly note that there is no difference I domain closure either. These points should be clarified.

---

## [Author Response]

Major issues that must be addressed, with additional experiments as necessary:Although this is broken down into separate points, 1 to 6 below, all of these issues are related to putting the activities of the relevant kinases on a more rigorous footing. We ask that a substantial response be prepared for these questions, but the authors should judge whether particular experiments can be done in a realistic timeframe or not. Note that these questions are important, so a simple response that none of the experiments can be done in a timely fashion will not be deemed a satisfactory response.1) The authors use MBP phosphorylation as readout for intrinsic activity, but as in their previous paper, they show that each ancestral kinase has a slightly different substrate specificity. Is it possible that the differences observed for MBP phosphorylation are due to difference in sequence specificity, rather than intrinsic activity?

This is a good point. To check this possibility more carefully, we added two experiments. First, we expanded our specificity arrays to include AncERK1-5 (see new Figure 1—figure supplement 3). There is not much change in specificity between AncERK1-5 and ERK2. We were unable to get good specificity array data for AncERK1-2 due to low activity. However, we believe it unlikely that a major specificity change would occur from AncERK1-5 to AncERK1-2 and then back again from AncERK1-2 to ERK2. Nevertheless, we also checked the relative activities of AncERK1-5 and AncERK1-2 with a second substrate, ELK1: see the new Figure 1—figure supplements 4A and 4B. We see the same drop in activity from AncERK1-5 to AncERK1-2 with this second substrate. Taken together, we are confident that our results indicate changes in activity without a major contribution from specificity changes.

2) In this assay, does MEK have any significant activity toward MBP? This should be measured as a control.

We tested this possibility and found that MEK has negligible toward MBP (Figure 1—figure supplement 2B). MEK is known to be highly specific for ERK, so we were happy to see this result. To further confirm that MEK can activate AncERK1-2, we co-expressed MEK and AncERK1-2, purified the ancestor away from MEK and found that this approach also led to increased AncERK1-2 activity (Figure 1—figure supplement 4C and 4D). Together, these experiments reinforce the conclusion that MEK can activate AncERK1-2.

3) The activity measurements done in the paper are endpoint assays of myelin basic protein phosphorylation. For at least the key proteins (ERK1, AncERK1-5, AncERK1-2), the authors should carry out kinetic experiments. This would allow them to determine whether the effects are driven by changes in k_cat_, K_M_, or a combination.

We undertook the suggested characterization and found that the decrease in activity is due to a combination of increased K_M_ and decreased k_cat_. Interestingly, the values for AncERK1-2 look very similar to reported values of unphosphorylated ERK2 and the values for AncERK1-5 look very similar to phosphorylated ERK2 (Prowse and Lew, JBC, 2000). We have added these data to a new Figure 1D.

Prowse, C.N., and Lew, J. (2001). Mechanism of Activation of ERK2 by Dual Phosphorylation. J Biol Chem 276, 99–103.

4) What are the phosphorylation states of each protein prior to the MBP phosphorylation reactions? It would be important to compare the activities of all proteins for the same phosphorylation states (nonphosphorylated versus doubly phosphorylated proteins). The activity data presented in Figure 1B represent the results of different mixtures of non- and phosphorylated kinases according to Figure 2B and C.For this, the quantification of phosphorylation for all samples need to be measured first, and then activity data collected for clean species (in Figure 2B and C, only a subset of proteins were characterized). Particularly AncERC1-5and AncRC1-2 are missing in respect to analysis of differential phosphorylation rates, the crucial proteins where the switch seem to happen in evolution.

We characterized the phosphorylation state of the TEY motif in AncERK1-5 and AncERK1-2. We added these data to Figure 2D. AncERK1-2 is almost completely unphosphorylated and AncERK1-5 is highly phosphorylated (consistent with the K_M_ and k_cat_ data reported above).

5) A distinction should be made throughout the manuscript between "intrinsic activity" and regulation by activation loop auto-phosphorylation. Differences in "intrinsic activities" would ideally be assessed by comparing the enzymatic activities of all kinases in the same phosphorylation states. In other words, how active are each of the unphosphorylated kinases, and how active are those same kinases after activation loop phosphorylation? What the authors are measuring in the beginning of the Results section, and in Figure 1B, appears to be the consequence of a difference in the ability to auto-phosphorylate and thus auto-activate, not necessarily a difference in intrinsic activity.

Thank you for pointing this out. We believe we are really looking at differences in phosphorylation state due to differential autophosphorylation (see above experiments). We have updated this language and interpretation in the text.

6) Related to these points, it would be informative to see raw data (phosphorimages) from the MBP phosphorylation reactions in Figure 1B as a figure supplement, rather than just a normalized quantification of endpoint intensities.

We have added examples of these raw data to the Figure 1—figure supplement 2A.

7) Finally, we ask that the amino acid sequences of all proteins (modern and resurrected) be included in the supplementary materials, and it would be of value to give the number of substitutions in the text. The full tree used for ancestral sequence reconstruction (ASR), including the species, needs to be included for reproducibility for the field. A supplementary figure that shows the probabilities of the nodes, and information of the posterior probabilities for the full sequences is essential to evaluate ASR.

We have added a supplementary figure with the posterior probabilities of the reconstructions for each of the kinases studied (Figure 1—figure supplement 1). We have also provided a link to all reconstruction data including the full trees, sequences for every ancestor and all probabilities: http://phylobot.com/350478581/msaprobs.PROTCATLG/ancestors

Other important issues that require response:1) The authors should avoid language that implies that the ancestral protein sequence is actually known. A sophisticated reader will understand what is meant, but we are concerned that the way that much of the paper is written a strong impression is given that the ancestral sequence is actually synthesized. This is not just a semantic point, because it may turn out that errors in the sequence reconstruction have led to misleading results. We do not consider this point to invalidate the conclusions of this study, but we ask that the language used to describe the "ancestral" sequences be changed throughout the manuscript. A few comments, given below, should help explain what we mean.The method of ancestral sequence reconstruction is a powerful approach to dissect molecular differences between closely-related biomolecules. However, the use of the dramatic term "resurrection" should be avoided throughout this manuscript, as it implies that we have explicit knowledge of the true protein sequences found in ancestral organisms, which is then "resurrected". The more common term, "reconstruction", is more appropriate, as it indicates that a sequence model of a plausible ancestral protein is being generated using the best available evidence. An example of a statement that implies that the ancestor is truly known, and that the authors should avoid is the following:"As part of this study, we observed that a deep ancestor of this group had relatively high intrinsic activity even in the absence of any activating kinase of or allosteric activators."In the beginning of the main section of the paper, the authors use the following phraseology, which correctly captures the inherent uncertainty in the "resurrected" sequences: "We synthesized the coding sequences for these inferred ancestors".

We have reworded the manuscript to better reflect these uncertainties and removed the term “resurrection”.

2) The manuscript is focused on the specific late evolution between ERK5 and ERK 1 and 2, and there is no information about the evolution of regulation towards CDKs, MAPKs, JNK, p38. Therefore the Introduction, Abstract and title should be modified to reflect the data from this manuscript (for instance: "Ancestral resurrection reveals mechanism of kinase regulatory evolution of ERK").

We have changed the title to reflect the focus on ERK.

3) To enlighten the reader about the evolutionary trajectory, key information to add would be how modern ERK5, ERK 7 that branch off the ancestors used in this study are behaving in respect to activities of their nonphosphorylated and phosphorylated activation loop states towards substrates, and their ability to autophosphorylate.

We have added language to indicate the high basal activity of ERK5 and ERK7 to the text.

4) What is the explanation of a much higher activity of AncERK1-2 with the two mutations in the presence of MEK relative to AncERK1-2 in the presence of MEK? Could the authors also include the data for the reverse experiment, AncERK1-5 M89Q 44Rdel, both in the absence and presence of MEK, in Figure 3?

This is a good question. We have added experiments to address this issue. Adding MEK to the AncERK1-2 Q89M ins44R doesn’t increase the kinase activity much. Adding MEK to AncERK1-2 increases activity ~6 fold, but this leads to < 10% of the activity of the active mutant. We believe that this inefficiency is due to the known high specificity of MEK for ERK1 and 2. We speculate that MEK doesn’t recognize AncERK1-2 as efficiently, accounting for the lower efficiency of activation of AncERK1-2.

We tried experiments adding MEK to AncERK1-5 (Figure 3—figure supplement 5) and saw only a slight potentiation. We also added MEK to AncERK1-5 M89Q del44R and again saw only an ~2-fold increase in activity. It is likely that AncERK1-5 is an even worse substrate for MEK than AncERK1-2. We have added these experiments to Figure 3—figure supplement 5.

5) 300 ns trajectories are very short MD simulations, and it is quite unlikely that refolding of the L16 loop coupled to domain closure is on that time scale. NMR data for ERK has suggested that these processes occur on the millisecond time-scale. Thus, the MD simulations are not convincing due to the discrepancy in time scale between the suggested conformational changes and the length of the simulations, and rather weaken the manuscript than strengthen it. The authors should consider carefully what information is contributed by the MD simulations, and then describe that with appropriate cautionary notes. Alternatively, remove the MD simulations from the manuscript.

We have greatly expanded our MD analysis using roughly three orders of magnitude more data. We performed extensive analyses of multiple (1,000) long MD simulations per sequence, generating ~500 microseconds aggregate simulation data for each sequence, and compared the WT and mutants. In total, we generated and analyzed a total of ~ 2 ms of aggregate simulation data.

We did not observe a sufficient number of very slow conformational rearrangements (e.g. DFG flips or large-scale domain motion) to be able to construct a Markov state model of the complete conformational dynamics. Therefore, we have taken care not to make such claims in the manuscript. However, this new far larger simulation dataset (> 3 orders of magnitude more than the previous experiment) is consistent with the leading edge of what has been done for biomolecular simulations of kinases, e.g., compare to the Scientific Reports paper from Sultan and Pande where they have 1.7 ms aggregate from the simulation of two kinase sequences (we have ~ 2 ms for 4 sequences):

Sultan, M.M., Denny, R.A., Unwalla, R., Lovering, F., and Pande, V.S. (2017). Millisecond dynamics of BTK reveal kinome-wide conformational plasticity within the apo kinase domain. Sci Rep *7*, 15604.

We are happy to report that the more active configuration of catalytic residues is maintained in these much more extensive data (Figure 5D-F). We can also clearly see synergy between the two mutations. Furthermore, each mutation gives increased flexibility at two points in the kinase (Figure A-C). While we can’t comment on domain closure and have therefore removed this language, we do see increased variance in domain angles (Figure 5G). These last data are presented with strong caveats. We believe these data provide starting points for future biochemical and simulation work and therefore are valuable to the study.

These new data were obtained in collaboration with John Chodera and Rafal Wiewiora at MSKCC, therefore they are now included as co-authors on the study. They have seen and approved the manuscript and are in agreement with their placement in the author list. They collaborated with Myvizhi Selvan, a new postdoc in the lab of Zeynep Gumus, therefore she is now added as an author. The work of Selvan Vatansever was removed from the manuscript following earlier criticisms and as a result she did not ultimately contribute to the data in the projects, therefore she has been removed as an author. All of these changes have been approved by the laboratories involved.

Should the MD simulations be retained, the authors should clarify what atoms are being used to measure the distances between residues in the analyses of MD simulations. In particular, for salt bridges, as in the one analyzed in Figures 5C, F, it would be most informative to look at distances between terminal or near-terminal atoms of the side chains (e.g. the terminal nitrogen of the lysine sidechain and the δ carbon or one of the oxygens of the glutamate residue).

We are indeed using the nearest neighbor heavy (non-hydrogen) atoms. We have added this language to the text.

6) The figures could be better arranged and assembled, it is hard to jump back and forth between gels and activity figures. Could all data be presented in a quantitative format (activity towards MBP and autophosphorylation)?

We have added the data in more quantitative format where possible.

7) The authors should include an LC/MS quantification of TEY motif phosphorylation for AncERK1-5 and AncERK1-2 (i.e., a similar analysis as was done in Figure 2C).

We have added these data in Figure 2D.

8) Figure 4E does not conclusively establish the mechanism of autophosphorylation; His6-ERK1 ins74N could still be undergoing intermolecular (trans) autophosphorylation. This point would need to be established by kinetic experiments, e.g. by measuring the concentration dependence of autophosphorylation.

We have added these data in Figure 4—figure supplement 4. The concentration dependence of autophosphorylation is consistent with intramolecular autophosphorylation.

9) Figure 6D: the authors should present a quantification of the Western blots (the ratio of phosphorylated ERK1 to total ERK1).

We have added this quantification.

10) The activation loops of the ancestral kinases differ in length and sequence. Notably, the ancestral CDK.MAPK kinase has a -3 Arg residue, for which this kinase still has some residual preference. If canonical sequence-specificity is not relevant because of a cis-autophosphorylation mechanism, it is still worth noting that the sequence of the activation loop could affect activation loop dynamics. The authors should discuss how the composition of activation loop itself might impact their findings. This is somewhat addressed by the chimera experiments but would still be worth mentioning.

We have added text that discusses this possibility in the section on chimeric kinases.

[Editors' note: further revisions were requested prior to acceptance, as described below.]

[…] 1) A major issue is that for AncERK, AncJPE, AncMAPK, AncCDK.MAPK (which are evolutionary further from the modern proteins), still no posterior probabilities are provided. They need to be shown, as previously asked by the review. In addition, no experiments were performed to test for their phosphorylation status, and no experiments were done on their modern counterparts. Only activity was measured for these ancestors (Figure 1A), but their relation in respect to modern CDK, MAPK, JNK, p38 is not discussed either in the manuscript.

We have added this discussion (subsection “Reconstruction of the lineage leading to ERK kinases”).

Without the proper analysis on those ancestors the authors may wish to remove them and to keep the focus on the ERK ancestors. The latter is the story of this manuscript, and this part is interesting and convincing.

We discuss the regulation of p38, ERK1-2 and CDKs in the Introduction as well as the outgroups DYRK and GSK groups of kinases.

We have added the posterior probabilities for all of these ancestors (Figure 1—figure supplement 1). We have also performed extensive additional mass spectrometry and have analyzed the phosphorylation status and added these results (new Figure 2C). Therefore, we have kept the deeper ancestors in the paper (additionally the specificity analysis in Figure 1—figure supplement 3 is valuable for future studies on the evolution of kinase specificity).

2) The manuscript still does not correctly describe the quantification of the percentages of non-phosphorylated, single and double phosphorylated (TxY motif) proteins used in the activity studies. Figure 2—figure supplement 1B is provided, but one cannot follow what is exactly shown, and how these data can be used for quantification of phosphorylation. In the text it is described as "more" or "less" phosphorylation. Since differences in autophosphorylation is the center of the findings of this manuscript, this needs to be quantified accurately. Are these full-length proteins, or peptides shown in this figure, and the results of what kind of experiment are shown? Even ERK 1 appear to have some phosphorylation. What does "modified TxY " mean?

We agree, these data were quite unclear before. We quantified tryptic peptides in this analysis. We have updated the Materials and methods section and manuscript to clarify these points (subsection “The higher basal activity of deep inferred ancestors is due to autophosphorylation of the TxY motif”, Figure 2 legend and Materials and methods). We have also reanalyzed all of our data and presented the fraction of pT-E-Y, T-E-pY, and pT-E-pY (e.g. where pT is phosphothreonine) for all experiments. We include the precise sequences and modifications of all peptides in Source data 1.

3) The part of the manuscript pertaining to Point No. 4 in the "Other Important Points" in the original review, still needs further clarification in the manuscript. If this result is puzzling and currently cannot be explained unambiguously with a model, it is important to articulate this for the reader.

The original question was: 4. What is the explanation of a much higher activity of AncERK1-2 with the two mutations in the presence of MEK relative to AncERK1-2 in the presence of MEK? Could the authors also include the data for the reverse experiment, AncERK1-5 M89Q 44Rdel, both in the absence and presence of MEK, in Figure 3?

Our reply was: This is a good question. We have added experiments to address this issue. Adding MEK to the AncERK1-2 Q89M ins44R doesn’t increase the kinase activity much. Adding MEK to AncERK1-2 increases activity ~6 fold, but this leads to < 10% of the activity of the active mutant. We believe that this inefficiency is due to the known high specificity of MEK for ERK1 and 2. We speculate that MEK doesn’t recognize AncERK1-2 as efficiently, accounting for the lower efficiency of activation of AncERK1-2.

We tried experiments adding MEK to AncERK1-5 (Figure 3—figure supplement 5) and saw only a slight potentiation. We also added MEK to AncERK1-5 M89Q del44R and again saw only an ~2-fold increase in activity. It is likely that AncERK1-5 is an even worse substrate for MEK than AncERK1-2. We have added these experiments to Figure 3—figure supplement 5. We have now added this discussion to the subsection “Shortening of linker loop and mutation of gatekeeper residue led to tight regulation in the ERK1-2 family”.

4) The MD simulations are done on ERK2 (Figure 5), but all the experiments in the manuscript are performed on ERK1 (Figure 4). Why is this?

This is because we have crystal structures of unphosphorylated ERK2 but not of ERK1. We chose to focus our analysis on unphosphorylated kinase as we decided the initial autophosphorylation would be most relevant. Furthermore, there was previous MD simulation work and mechanistic suggestions on ERK2 autophosphorylation (Barr, et al., 2011 and Emrick, et al., 2006), hence we decided it would be most valuable to extend this research direction on the same system.

We have performed biochemical experiments with ERK2 and obtain similar results as with ERK1, i.e. the same gatekeeper and insertion mutations lead to increased activity due to higher basal phosphorylation. Therefore, we believe that results that we obtain with the ERK2 structure will be relevant to both ERK1 and ERK2. We have added the ERK2 experiments to Figure 5—figure supplement 1. We have also now explained our reasoning in the manuscript (subsection “Molecular dynamics simulations reveal increased flexibility at the glycine rich loop and loop 16, and a higher probability of an active organization of catalytic residues in mutant kinases”, first paragraph).

There is difficulty in seeing how the MD figures provided now help explain why the mutants of ERK1 autophosphorylate the TxY motive in cis much more efficiently than wtERK1. In Figure 5 all proteins have a bimodal distribution for the plotted distances (which are slightly altered). If one of these two substates is interpreted to be the active conformation, one would not expect the experimental result of a big difference in phosphorylation state between these proteins. The authors correctly note that there is no difference I domain closure either. These points should be clarified.

We have performed the MD simulations to explore some of the structural changes influenced by the mutants and suggest hypotheses for further mutational studies. While these simulations were extensive and long (2.05 ms in aggregate from 4,000 MD simulations for four systems), they were still not long enough to sample and capture some of the conformational changes that occur on a millisecond scale, such as domain closure. Hence, the data clearly do not describe the true (global) equilibria of the systems, rather it is informative about the neighborhood (local equilibrium) of the inactive conformation. While not offering definitive explanations as to the full autophosphorylation activation pathway, the data show the relaxation processes of the mutated systems that would eventually lead to the global equilibrium. Hence, it is not surprising that we observe only small changes to the conformational ensembles at these limited timescales, and with more sampling we expect the differences to keep increasing until we have described the full activation mechanism leading to the experimental magnitude of change in activity. However, knowledge of the initial stages of this relaxation process towards the global equilibrium, particularly the patterns of residue-residue contact disruptions, would provide enough information to make informed hypotheses about mutations that can further tweak the behavior of the protein.

To maximize the understanding of the structural changes seen in the current datasets, we have now developed Markov state models (MSM) trained on the inter-residue distances corresponding to contacts most relevant to distinguishing the conformational ensembles of the four protein systems from each other. MSMs help integrate information from multiple simulations to reveal the system’s kinetically distinct macrostates, whose populations can be perturbed by mutations. The MSMs revealed how different structural changes caused by the mutants are kinetically coupled with each other. Again, these insights will allow for generation of further mutational hypotheses to further study structural aspects of the autophosphorylation activation process.

In addition to the mutants promoting formation of active-like conformations, cis-autophosphorylation requires that the phosphorylation sites be able to intramolecularly reach the active site – a mechanism of activation lip destabilization leading to that has been previously proposed (Emrick et al., 2006). We have analyzed our data according to this proposed mechanism, and found the presence of small populations of conformations where the phosphorylation site tyrosine can access the active site aspartate in all systems. We did not observe other changes proposed by the mechanism, instead we analyzed the patterns of contact changes in our simulations at the activation lip and indicate crucial residues for future mutational studies.

We have updated the text to indicate the observations and limitations of our current MD simulation. The results of these simulations will be used in future studies as seeds for adaptive sampling to generate MSMs describing all relevant activation steps involved, and to generate first guesses at reaction coordinates for enhanced sampling. We note that a description of the full activation mechanism would be a significant research project in itself, and is far outside the scope of this work.